# A blockchain-based multi-authority hierarchical attribute encrypted data sharing scheme in the Internet of Medical Things

Hao Yuan[1,2], Guofang Dong[1,2]*, Leilei Zhao[1,2]

**1** School of Electrical and Information Technology, Yunnan Minzu University, Kunming, China, **2** Yunnan Key Laboratory of Unmanned Autonomous System, Yunnan Minzu University, Kunming, China

* dgfynmzdx@163.com

## Abstract

With the rapid development of the Internet of Medical Things (IoMT), the secure and efficient sharing of massive amounts of sensitive medical data has become a core challenge. Addressing the limitations of existing Ciphertext-Policy Attribute-Based Encryption (CP-ABE) schemes, such as the lack of data source authentication, computational redundancy, and single-point-of-failure risks when handling hierarchical data, this paper proposes a blockchain-based multi-authority hierarchical attribute-based encryption scheme. First, the scheme integrates a Distributed Key Generation (DKG) protocol and combines threshold BLS signature technology to establish a collaborative authentication mechanism, thereby enhancing the verification of data source authenticity. Additionally, a dynamic update mechanism ensures the long-term security of collaborative key management. Second, the scheme optimizes the encryption logic for structured data by constructing a hierarchical access tree, and introduces a multi-authority collaboration mechanism and proxy re-encryption (PRE) technology to mitigate single-point-of-failure risks and enable efficient user permission revocation. Security analysis demonstrates that the scheme is resistant to chosen-plaintext attacks (IND-CPA) and collusion attacks by authorities under standard models. Meanwhile, the DKG protocol has been proven to satisfy validity, robustness, confidentiality, and resistance to Sybil attacks. Performance evaluation indicates that the CP-ABE algorithm in this scheme outperforms existing solutions in terms of computational and storage overhead. In large-scale testing on a 100-node Hyperledger Fabric environment, the system achieved a consensus latency of approximately 280 ms and a key update propagation delay of 1.52 s, validating the feasibility of deploying this solution in real-world IoMT environments with limited resources and certain real-time requirements.

**Data availability statement:** All relevant data are within the manuscript and its Supporting information files.

**Funding:** The author(s) received no specific funding for this work.

**Competing interests:** The authors have declared that no competing interests exist.

## 1. Introduction

With the exponential growth of the IoMT, massive numbers of wearable sensors and remote monitoring devices are driving the transformation of healthcare services toward real-time, intelligent capabilities. While this trend enhances personalized medical care, it also poses significant challenges for the secure sharing and granular governance of medical data [1]. Given the highly sensitive nature of medical data and the need to comply with regulations such as the Health Insurance Portability and Accountability Act (HIPAA) and the General Data Protection Regulation (GDPR), ABE is regarded as a key cryptographic solution for achieving fine-grained data authorization [2]. This technology primarily comprises two branches: Key Policy Attribute-Based Encryption (KP-ABE) [3], and CP-ABE [4]. In comparison, CP-ABE allows data owners to autonomously define access policies and embed them within the ciphertext, aligning more closely with patients' stringent privacy controls in healthcare settings. As a result, CP-ABE is widely recognized as the core tool for safeguarding the privacy of IoMT data [5–8]. However, in practical IoMT applications, existing CP-ABE schemes still face the following critical technical bottlenecks when deployed in distributed environments and resource-constrained devices (as illustrated in Fig 1).

Firstly, data sources lack lightweight mechanisms for verifying authenticity. As shown in the upper half of Fig 1, IoMT terminals are typically deployed in uncontrolled physical environments. Attackers can easily hijack or tamper with terminal devices to replace genuine physiological data $M_1$ with fabricated data $M'_1$. Traditional CP-ABE schemes often focus solely on confidentiality protection during static storage, neglecting the legitimacy of the encryption initiator's identity. If falsified data bypass verification and enter the system directly, they will mislead subsequent clinical decision support. Therefore, integrating a lightweight decentralized traceability mechanism into the encryption process constitutes the first line of defense for securing IoMT systems.

Secondly, traditional CP-ABE encryption schemes struggle to accommodate the hierarchical structure of medical data, leading to severe efficiency bottlenecks. In IoMT scenarios, data inherently exhibits logical interconnections. As shown in the lower half of Fig 1, the detailed surgical records $M_1$ and the routine vital sign monitoring data $M_2$ for the same patient correspond to the access policies $p_1$ and $p_2$, respectively. Existing solutions typically require separate encryption processes for each data file, resulting in substantial redundant ciphertext. For IoMT devices constrained by computational power and storage capacity, this non-hierarchical approach not only causes severe storage waste but also significantly increases data processing latency. Leveraging hierarchical data relationships to achieve single encryption with multi-level authorization is key to enhancing IoMT sharing efficiency.

Finally, trust models based on a single centralized authorization face single-point-of-failure risks. In complex IoMT management architectures, if key generation and distribution rely entirely on a single authority, an attack or failure at this core node would collapse the entire system's trust boundary. Furthermore, as healthcare personnel roles dynamically change, achieving low-overhead attribute revocation and

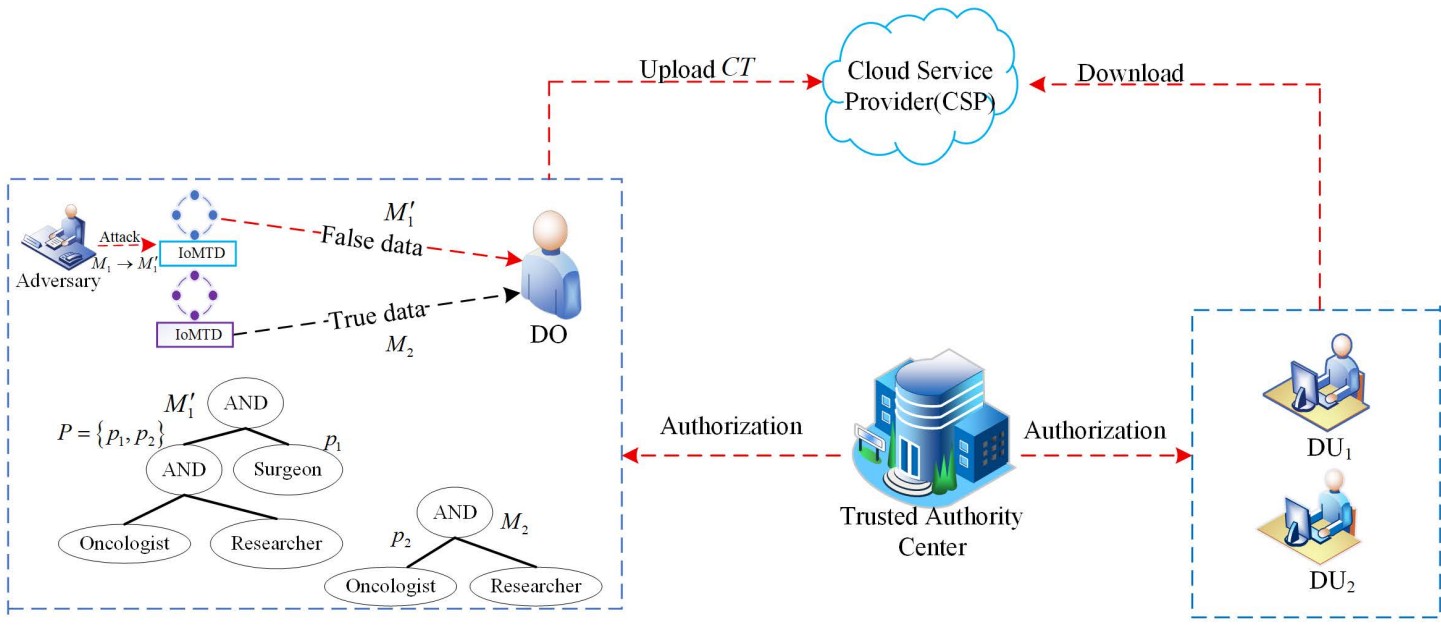

**Fig 1. Shared encrypted data process.**

key rotation in distributed scenarios involving multiple authorities remains a major challenge for existing solutions, particularly in terms of system scalability and robustness.

To address these challenges, we propose BMHADS, a blockchain-based multi-authority hierarchical CP-ABE framework. To balance security and efficiency in IoMT environments, the scheme is instantiated on Type-3 pairing-friendly curves (e.g., BLS12−381). Compared to traditional Type-1 pairings, Type-3 curves yield shorter ciphertexts and superior computational efficiency at the same security level, making them ideal for resource-constrained devices. Our main contributions are as follows:

(1) A collaborative authentication mechanism that integrates blockchain-based DKG with BLS threshold signatures has been proposed. This mechanism uses auxiliary nodes for collaborative verification, transforming centralized authentication into a distributed consensus process that ensures data authenticity and decentralized fault tolerance. Furthermore, the mechanism incorporates Proactive Secret Sharing (PSS) technology to support periodic key updates, enhancing the long-term security of system credentials without altering public keys.

(2) This paper proposes a hierarchical multi-authority CP-ABE scheme optimized using Type-3 curves. The scheme utilizes a hierarchical access tree to encrypt multiple associated files in a single operation, thereby eliminating computational redundancy. At the same time, by having multiple authorities independently issue key fragments and aggregate them to generate the user's private key, the scheme fundamentally alleviates the key escrow problem. Thanks to the compact representation of group elements in Type-3 curves, the scheme reduces storage and computational overhead, thereby ensuring its feasibility for deployment on resource-constrained IoMT sensors.

(3) This paper proposes a periodic revocation mechanism based on PRE to address the efficiency bottleneck associated with user revocation in dynamic environments. By strategically offloading computationally intensive key update and ciphertext re-encryption tasks to healthcare cloud service providers, this mechanism enhances system scalability and reduces local overhead.

(4) Security analysis demonstrates that the proposed BMHADS scheme achieves IND-CPA security and effectively resists collusion attacks. Furthermore, the validity, robustness, confidentiality, and resistance to Sybil attacks of the DKG protocol have been further verified, providing a trusted environment for the sharing of sensitive medical data. Experimental results indicate that the proposed BMHADS scheme outperforms existing schemes in terms of storage and computational overhead. Furthermore, implementation and testing were conducted on a Hyperledger Fabric platform comprising 100 nodes. The system maintained acceptable blockchain storage overhead and low consensus latency, validating the feasibility of deploying this scheme in real-world IoMT scenarios.

**Organization of the Paper.** The organizational structure of the BMHADS proposal is as follows: First, Sections 2 and 3 provide a review of the current state of research and introduce the necessary theoretical background. Subsequently, Section 4 defines the proposal's system architecture and security model. Building on this foundation, Section 5 details the specific design aspects of the BMHADS proposal. The security validation and performance evaluation of the proposal are discussed in Sections 6 and 7, respectively. Finally, Section 8 summarizes the entire work and provides a look ahead to future research.

## 2. Related work

To circumvent the inherent risks of single points of failure and key custody in traditional CP-ABE architectures, multi-authority collaboration mechanisms have become a key focus in academic research. Chase et al. [9] pioneered a multi-authority framework that leverages global identity identifiers to achieve cross-institutional anti-collusion properties. However, this approach remains highly dependent on central authorities and faces risks of user privacy leakage. Building upon this foundation, functional optimizations tailored for specific scenarios have subsequently emerged. Duan et al. [10] combined PRE to enable efficient authorization delegation across chains, empowering data owners to update access policies dynamically. Zhao et al. [11] focused on enhancing terminal performance by proposing an online/offline multi-authority scheme supporting Linear Secret Sharing Scheme (LSSS) policy hiding, effectively reducing computational overhead on mobile devices. However, while multi-authority architectures ensure flexibility, they pose stringent challenges to the security and overhead of dynamic revocation mechanisms. To address this, Liu et al. [12] designed a multi-authority framework that enables instant revocation by removing central authorities and incorporating server-side key deletion. However, its security heavily relies on cloud integrity. Subsequently, Varri et al. [13] introduced a dual-authority collaborative architecture that supports identity tracing and achieves indirect revocation through evolutionary key generation. However, its substantial communication overhead severely limits scalability when handling large numbers of terminals.

Beyond functional and revocation management, researchers have also focused on deepening the theoretical boundaries of decentralization through mathematical frameworks. A landmark contribution in cryptography was achieved by Lewko et al. [14], who proposed the first fully decentralized multi-authority scheme requiring no global coordination. Building on this foundation, Liang et al. [15] introduced an anonymous distribution protocol that provides dual privacy concealment for both user identities and access policies. Subsequently, Qian et al. [16] proposed a multi-authority key-generation scheme in which authorities use shared random seeds to collaboratively generate key components. Combined with PRE techniques, this approach achieved a favorable trade-off between security and revocation efficiency. To address the complex hierarchical structure of medical records, hierarchical encryption techniques were introduced to enhance efficiency further. Bobba et al. [17] employed recursive set construction to form hierarchical properties. Wang et al. [18] and Xiao et al. [19] achieved ciphertext component reuse by integrating multi-level access frameworks, significantly reducing redundancy overhead. In the field of electronic health record and personal health record sharing, Guo et al. [20] and Roy et al. [21] implemented a layered scheme combining one-time encryption with multi-level authorization. Unlike layered schemes based on Type-1 symmetric pairs $(\mathbb{G}_1 = \mathbb{G}_2)$ [20,21], this study employs Type-3 asymmetric pairs. This choice reflects a core design trade-off: although fine-grained parameter partitioning across heterogeneous groups increases

implementation complexity, it fundamentally resolves the parameter bloat issue at high security levels. Compared to the 3,072-bit characteristic width required by Type-1 curves to achieve a 128-bit security level, Type-3 curves require only 381 bits to withstand attacks of the same severity, making them better suited to the resource constraints of IoMT endpoints. Furthermore, most of these layered methods focus on post-storage static confidentiality while neglecting data source authentication at the perception layer, leaving the system highly vulnerable to data injection attacks at the source stage.

Ensuring data reliability throughout its entire lifecycle is another core requirement in IoMT scenarios. Traditional auditing solutions [22–24] heavily rely on third-party intermediaries, posing risks of single points of failure. To address this, Liang et al. [25] and Tian et al. [26] attempted to build decentralized auditing mechanisms using blockchain technology. However, existing blockchain-assisted solutions still fall short in terms of security depth. While Yu et al.'s [27] approach enhanced management capabilities, it lacked integrity auditing. Lee et al.'s [28] solution reinstated auditing functionality but neglected upstream authenticity verification. Addressing this gap in data source validation, researchers pioneered a technical pathway from identity matching to collaborative authentication. Ateniese et al. [29] pioneered the secret handshake protocol. Subsequently, Xu et al. [30] and the latest Yao et al. [31] proposals ensured participant authenticity through bidirectional attribute matching. However, such passive authentication models struggle to resist false data injection after terminal physical hijacking. Although Qi et al. [32] and Zhang et al. [33] attempted to address the aforementioned shortcomings by using aggregated signatures, their models typically follow a collect-then-compress logic, combining signatures from multiple sources for verification. This approach makes the system highly vulnerable to single points of failure when the credentials of a specific device are compromised. To address this issue, the collaborative authentication mechanism proposed in this paper achieves substantial improvement over aggregate signature methods by integrating DKG-based $(t, P)$ threshold signatures. Unlike traditional signature aggregation, this mechanism ensures that a valid data-source signature can be generated only when at least $t$ assistant nodes reach consensus. This design provides decentralized fault tolerance and guarantees that the authenticity of the medical data stream remains unforgeable even if up to $t-1$ assistant nodes are compromised—a security feature not present in the literature [32,33].

In this study, although the hierarchical access tree mechanism and the attribute revocation mechanism are borrowed from schemes [20,21] and scheme [16], respectively, the proposed BMHADS scheme achieves a fundamental evolution in its underlying mathematical framework. By adopting Type-3 asymmetric pairing, this scheme eliminates the parameter redundancy issue present in traditional Type-1 schemes from an algebraic perspective, thereby making it more suitable for resource-constrained IoMT scenarios. Furthermore, building upon the aggregated signature approach in [32,33], we introduce a threshold signature mechanism based on the DKG protocol. This improvement facilitates a shift from simple signature aggregation to decentralized consensus verification, thereby enabling robust data-source verification for hierarchical CP-ABE schemes in IoMT environments.

## 3. Preliminaries

### 3.1. Bilinear mapping

A cryptographic bilinear map operates over cyclic groups $\mathbb{G}_1$, $\mathbb{G}_2$ and $\hat{\mathbb{G}}_T$ of prime order $r$, defined as a function $\hat{e} : \mathbb{G}_1 \times \mathbb{G}_2 \to \hat{\mathbb{G}}_T$ satisfying:

(1) Bilinearity: $\hat{e}\left(g_1^{\varrho_1}, g_2^{\varrho_2}\right) = \hat{e}(g_1, g_2)^{\varrho_1 \varrho_2}$ For any $g_1 \in \mathbb{G}_1, g_2 \in \mathbb{G}_2$, and $\varrho_1, \varrho_2 \in \mathbb{Z}_r^*$.

(2) Non-degeneracy: $\hat{e}(g_1, g_2) \neq 1$ if $g_1$ and $g_2$ are generators.

(3) Computability: Effective algorithms exist to compute $\hat{e}(g_1, g_2)$.

The proposed BMHADS scheme is instantiated using Type-3 pairings, where security is underpinned by the Elliptic Curve Discrete Logarithm Problem (ECDLP) within $\mathbb{G}_1$ and $\mathbb{G}_2$, as well as the Finite Field Discrete Logarithm Problem (FFDLP) in the target group $\hat{\mathbb{G}}_T$. Due to the absence of efficient computable isomorphisms between $\mathbb{G}_1$ and $\mathbb{G}_2$ in Type-3

structures, security degradation associated with group homomorphisms is effectively eliminated. This structural advantage provides a more robust mathematical foundation for implementing multi-authority hierarchical CP-ABE and DKG-based collaborative authentication.

## 3.2. Decisional Bilinear Diffie-Hellman (DBDH) assumption

Given a valid Type-3 pairing defined over a parameter set $\left(\mathbb{G}_1, \mathbb{G}_2, \hat{\mathbb{G}}_T, g_1, g_2, r\right)$, the DBDH hypothesis is said to hold if there is no probabilistically polynomial-time (PPT) algorithm capable of distinguishing, with non-negligible advantage, a DBDH tuple $\left(g_1^a, g_1^b, g_1^c, g_2^a, g_2^b, g_2^c, \hat{e}\left(g_1, g_2\right)^{abc}\right)$ from a random tuple $\left(g_1^a, g_1^b, g_1^c, g_2^a, g_2^b, g_2^c, \mathcal{Z}\right)$. Here, $a, b, c \in \mathbb{Z}_r^*$, $g_1 \in \mathbb{G}_1$, $g_2 \in \mathbb{G}_2$, and $\mathcal{Z} \in \hat{\mathbb{G}}_T$ are uniformly randomly selected.

## 3.3. Hierarchical access tree

As shown in Fig 2, $\tilde{\mathbb{T}}$ denotes a hierarchical access structure that integrates diverse access policies and security levels into a unified framework. This structure is partitioned into $l$ distinct access levels, with the root node designated as $R$. To facilitate the formal analysis of this hierarchical framework, the following nomenclature and characteristics are introduced:

**Non-leaf nodes:** These represent threshold gates (e.g., *AND*, *OR*, or *n*-of-*m* gates, where $1 \le n \le m$). For instance, nodes $A$ and $B$ in Fig 2 are non-leaf nodes.

**Leaf nodes:** These represent specific attributes, such as nodes $C$ and $G$.

$num_{(x,y)}$: Denotes the number of child nodes of node $(x,y)$. For example, $num_R = 2$ and $num_B = 3$.

$(x_m, y_m)$: The access tree $\tilde{\mathbb{T}}$ is hierarchically organized into $l$ levels. Let $(x_m, y_m)$ denote the coordinates of a node situated at the $m$-th level, where $1 \le m \le l$. Specifically, the root node $R$ resides at level $m = 1$ and is represented as $(x_1, y_1)$. As $m$ increases, the depth of the node within the tree increases accordingly.

$th_{(x,y)}$: The threshold value associated with node $(x,y)$, where $1 \le th_{(x,y)} \le num_{(x,y)}$. It defines the logical behavior of non-leaf nodes: it functions as an *OR* gate if $th_{(x,y)} = 1$, and as an *AND* gate if $th_{(x,y)} = num_{(x,y)}$.

**Transmission node:** A node $(x,y)$ is classified as a transmission node if at least one of its children is a non-leaf node (threshold gate). As illustrated in Fig 2, node $A$ serves as a transmission node.

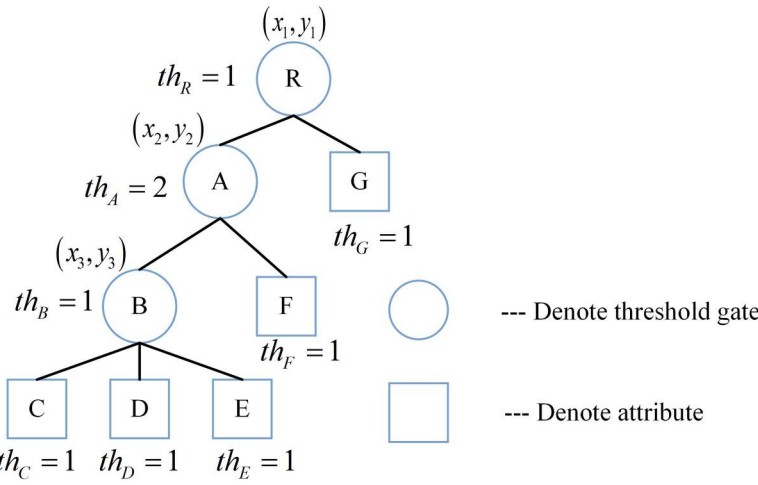

**Fig 2. Access tree with three hierarchies.**

***TN – SET(x, y)*:** The ensemble of threshold gates among the offspring of the transmission node $(x,y)$ within $\tilde{\mathbb{T}}$, i.e., $TN - SET(x, y) = \{ch_1, \ldots, ch_{j'}\}$, for instance, $TN - SET(A) = \{B\}$.

$\tilde{\mathbb{T}}_{(x,y)}$**:** Denotes a subtree of $\tilde{\mathbb{T}}$, with node $(x,y)$ its apex. $\tilde{\mathbb{T}}_{(x,y)}(S)$ determines if attribute set $S$ conforms to access tree $\tilde{\mathbb{T}}$. Additionally, $\tilde{\mathbb{T}}$ is computed recursively: For a leaf node $(x,y)$, $\tilde{\mathbb{T}}_{(x,y)}(S) = 1$ precisely when $(x, y) \in S$. For non-leaf nodes $(x,y)$, $\tilde{\mathbb{T}}_{(x,y)}(S) = 1$ if and only if at least $th_{(x,y)}$ child nodes satisfy the condition.

## 4. System framework

This section provides a detailed discussion of the system model, system threat model and assumptions, security model, and related considerations for the proposed BMHADS approach.

### 4.1. System model

Fig 3 displays the system model of the BMHADS method in the IoMT, and the model contains: Internet of Medical Things Device (IoMTD), Data Owner (DO), Healthcare Cloud Service Provider (H-CSP), Assistant Nodes (ANs) Attribute Authorities (AAs), Data User(DU), Blockchain (BC), Central Authority (CA).

**IoMTD**: IoMTD includes implantable sensors, everyday wearables, etc., which can collect a variety of medical data, sign the collected medical data using their BLS private key, and securely send it to an assistant periodically.

**ANs**: ANs are the core computational entities of this protocol, each with a unique identity and sufficient computing resources. They obtain participation qualifications by registering blockchain addresses and encrypted public keys, and by

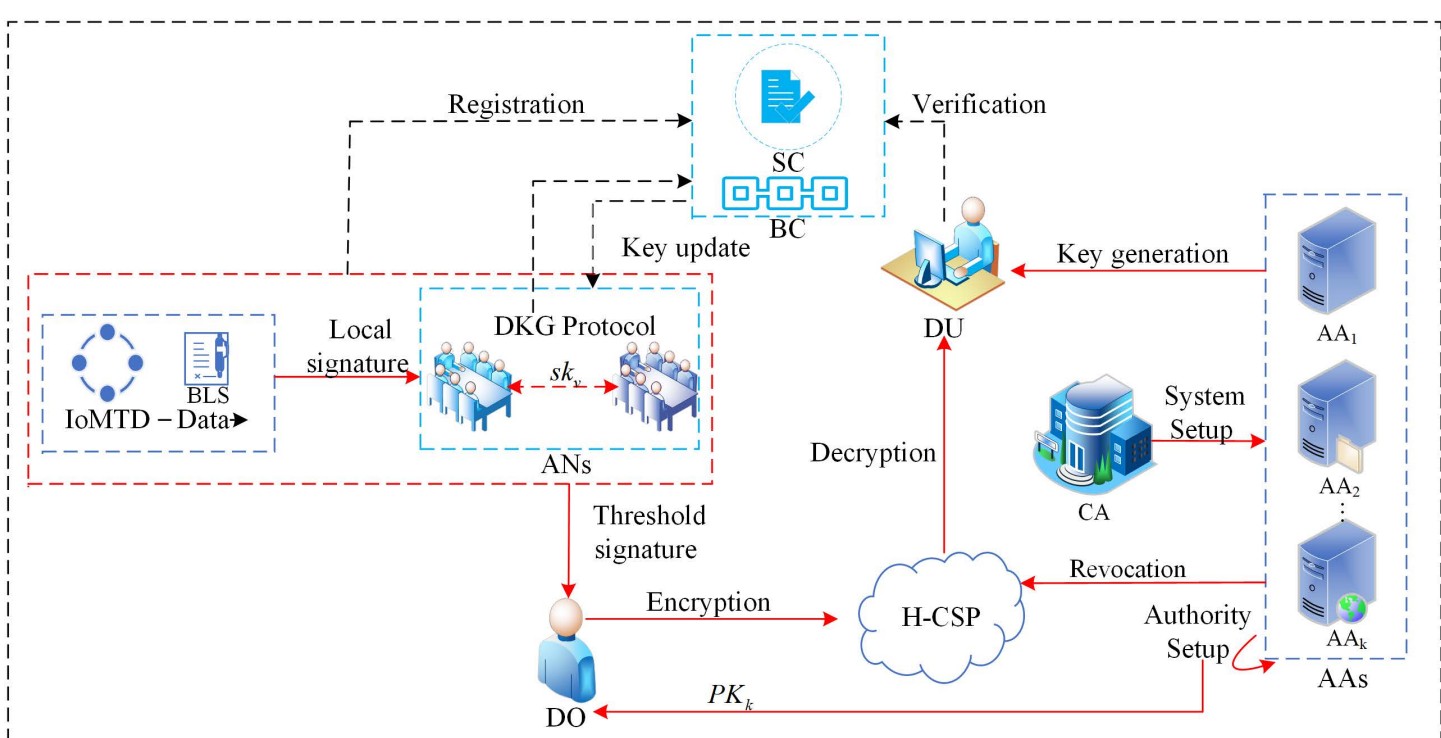

**Fig 3. System model.** The block labeled "DKG Protocol" encapsulates the phases of initialization, commitment and share distribution, complaint, and key reconstruction. For a detailed illustration of the protocol interactions, refer to Fig 4.

staking funds. They collaboratively build a decentralized trust infrastructure and execute distributed cryptographic protocols to achieve reliable data source verification.

**DO**: DO is the owner of the IoMTD, can manage one or more IoMTDs, and is responsible for collecting threshold signatures from assistants. DO packages the transactions to the blockchain, defines the corresponding access policies for the raw data, and encrypts and transfers them to the H-CSP.

**H-CSP**: In our scheme, the H-CSP serves as a semi-trusted third party responsible for storing medical data collected via IoMTD and is curious about sensitive data.

**AAs**: Multi-authority AAs are responsible for managing user attribute sets, issuing attribute keys and data owner public keys, communicating with users using anonymous key exchange protocols, and switching to a standalone mode of operation after configuring pseudo-random functions (PRFs) through interactions among AAs during the initialization phase.

**DU**: The DU first uses blockchain credentials to verify the authenticity of the data source. Upon success, the DU requests, downloads, and decrypts the data from the H-CSP. If the DU's attribute set satisfies the access structure for part or all of the encrypted data, the corresponding private key allows decryption of the ciphertext to obtain the medical data.

**CA**: The CA produces the system's public parameters and does not carry any keys or perform key-generation algorithms for other entities.

**BC**: The foundation of this agreement is a permissioned consortium blockchain with Byzantine Fault Tolerance (BFT) consensus operated by ANs; the on-chain smart contracts (SC) serve as a trustless arbitrator, responsible for node identity management, stake escrow, and enforcement of penalty rules.

### 4.2. Threat model and assumptions

To ensure the reliability of data sharing and the resistance against Sybil attacks, we establish the following assumptions regarding the blockchain environment and the adversary's economic behavior:

#### 4.2.1. Blockchain and smart contract trust assumptions.

1. **Consistency of the state tree:** BFT consensus ensures the global state tree (which records all nodes' registration addresses, public keys, stake amounts, penalty records, etc.) is immutable and consistent across honest nodes, preventing any adversary from forging historical data or identity records through local ledger tampering.

2. **Honest arbitrator:** The SC deployed on the blockchain is regarded as an honest arbitrator. Its execution logic, including identity activation, stake escrow, and penalty enforcement, is strictly enforced by the decentralized consensus protocol. An adversary $\mathcal{A}$ cannot bypass the contract logic to activate unauthorized identities unless they satisfy the necessary conditions, such as submitting a valid stake or completing identity verification.

#### 4.2.2. Adversary financial capability.

1. **Financial rationality:** The adversary $\mathcal{A}$ is assumed to be financially rational, meaning its primary objective is to maximize illicit profit. An attack will only be initiated if the potential gain (e.g., unauthorized access to sensitive medical data) exceeds the total cost of the attack (e.g., hardware costs and forfeited stakes).

2. **Bounded resources and economic stake:** The financial resources of $\mathcal{A}$ are bounded. To prevent Sybil attacks where an attacker creates multiple identities to gain control over the DKG process, the system enforces a minimum stake threshold $E'_{min}$ during node registration:

$$E'_{min} = (f + 1) \cdot \Omega \tag{1}$$

where $f$ denotes the maximum number of Byzantine nodes the system can tolerate, and $\Omega$ represents the preset non-negative minimum stake per node. To ensure economic security, $E'_{min}$ must be significantly higher than the potential illegal gains from accessing medical data, thereby making a large-scale Sybil attack economically unfeasible.

## 4.3. Security model

We define the security of our BMHADS scheme against IND-CPA under the selective security model based on the DBDH assumption. The security game takes place between a PPT adversary $\mathcal{A}$ and a challenger $\mathcal{C}$, consisting of the following stages:

**System Setup:** $\mathcal{A}$ determines and submits to $\mathcal{C}$ the challenge access structure $\tilde{\mathbb{T}}^*$ that it intends to attack, which will embed the encryption in plaintext form, meaning this scheme does not provide a strategy hiding feature. A collection of corrupt authority institutions that it wishes to control is defined as $\mathcal{V}_A \subset \{A_1, A_2, \ldots, A_N\}$, and this collection must satisfy $|\mathcal{V}_A| \leq N - 2$, meaning at least two authority institutions are honest and not controlled by $\mathcal{A}$. $\mathcal{A}$ will acquire the private keys of all authority institutions in the collection in subsequent stages. In response, $\mathcal{C}$ executes the system setup algorithm, generates the public parameters, and sends them to $\mathcal{A}$.

**Authority Setup:** For each corrupt authority $A_k \in \mathcal{V}_A$, $\mathcal{C}$ also provides its corresponding public-private key pair $(PK_k, SK_k)$ to $\mathcal{A}$. For honest authorities, $\mathcal{C}$ only provides the corresponding public key $PK_k$.

**QueryPhase 1:** $\mathcal{A}$ adaptively submits a series of attribute sets $\{S_1, S_2, \ldots, S_o\}$ to request user key generation. For each attribute set $S$, $\mathcal{C}$ runs the *KeyGen* algorithm, but only returns the corresponding private key if $S$ does not satisfy the challenge access structure $\tilde{\mathbb{T}}^*$.

**Challenge:** $\mathcal{A}$ submits two messages, $M_0$ and $M_1$, of equal length. $\mathcal{C}$ randomly flips a coin $\xi \in \{0, 1\}$, encrypts $M_\xi$ using the algorithm $Enc\left(\tilde{\mathbb{T}}^*, M_\xi, PK_k\right)$ under the challenge structure, and returns the resulting challenge ciphertext $CT^*$ to $\mathcal{A}$.

**QueryPhase 2:** $\mathcal{A}$ can continue to adaptively query more sets of attributes $S_{o+1}, \ldots, S_{\bar{o}}$ for the key, with the restriction that these sets must not satisfy $\tilde{\mathbb{T}}^*$.

**Guess:** $\mathcal{A}$ outputs a guess value $\xi'$ for $\xi$. If $\xi' = \xi$, then the adversary wins the game. The advantage of the adversary $\mathcal{A}$ in this secure game is defined as $Adv_\mathcal{A}(1^\kappa) = \left| Pr(\xi' = \xi) - \frac{1}{2} \right|$.

**Definition 1:** Suppose the system has at least two honest authoritative institutions, and the advantage of any adversary $\mathcal{A}$ who operates in polynomial time in the aforementioned security game is negligible. In that case, the BMHADS scheme is secure against IND-CPA and is resistant to collusion among authorities according to the selective model.

During the execution phase of the DKG protocol, a computationally bounded adaptive adversary exists that can control at most $f$ nodes. These controlled nodes can exhibit arbitrary Byzantine behaviour, including, but not limited to, sending incorrect messages, refusing to respond to protocol requests, or engaging in other collusive actions. To ensure the protocol's availability, honest nodes must constitute an absolute majority. Assume there are a total of $P$ assistant nodes in the system, with $f$ being the number of malicious nodes and $H = P - f$ being the number of honest nodes. According to the requirements of Byzantine fault-tolerant consensus mechanisms, the total number of nodes must satisfy $P \geq 3f + 1$; thus, the number of honest nodes $H \geq 2f + 1$. In the key reconstruction phase, the protocol sets a threshold $t = 2f + 1$, meaning that collecting only $t$ valid shares is sufficient to reconstruct the system's master private key. In addition to satisfying IND-CPA security and resistance to collusion by authorities, the BMHADS scheme proposed in this paper should also achieve the following security properties:

1. **Validity:** ensures the successful execution of the DKG process through three key requirements. Firstly, any $t$ honest nodes can cooperatively reconstruct the master private key and aggregate a valid signature. Secondly, all honest participants must maintain a single, consistent protocol public key $PK_{group}$. Finally, the local private key share $sk_v$ of each honest node must remain valid and usable for generating threshold signatures that are verifiable by the global public key $PK_{group}$.

2. **Robustness:** Even if there are $f$ malicious nodes, as long as the total number of nodes $P$ in the system satisfies $P \geq 3f + 1$, all honest nodes can output valid group public keys $PK_{group}$ and their respective valid private key shares $sk_v$, thereby enabling participation in subsequent threshold signature operations.

3. **Confidentiality:** ensures that attackers gain no computational advantage in obtaining the master private key $SK_{group}$, thereby maintaining static secrecy against any colluding group and providing forward secrecy by preventing expired key shares obtained after time $T$ from compromising historical data.

4. **Sybil attack resistance:** ensures that the cost of identity acquisition increases superlinearly with the number of nodes $P$, rendering large-scale attacks economically unfeasible as established in the threat model (Section 4.2).

## 4.4. Discussion on security models

The proposed BMHADS scheme has demonstrated selective security under the DBDH difficulty assumption. The following discussion addresses the selection and application of security models:

1. **Feasibility of full security:** The full adaptive security model allows an adversary to dynamically choose their challenge access structure after observing the public parameters and performing multiple secret-key queries, providing a more realistic simulation of high-intensity attack environments. However, in the field of ABE, particularly in complex scenarios involving multi-authority and hierarchical verification, achieving adaptive security often requires Dual System Encryption techniques. Such techniques often lead to a significant increase in the number of ciphertext components and require more complex bilinear pairing operations. Given the stringent requirements for real-time processing and low computational overhead in IoMT and PHR systems, our scheme adopts the selective security model to strike an optimal balance between functional integrity and computational efficiency.

2. **Practical limitations and justification:** The primary limitation of the selective security model is the requirement that the adversary commit to the target access structure before system initialization, which limits the adversary's ability to adjust attack strategies dynamically. However, in practical distributed medical scenarios, user attribute permissions (e.g., chief physicians, researchers, or head nurses) and the system's access policies are usually determined by a stable organizational hierarchy. These structures are relatively static and do not undergo fundamental changes within milliseconds. Therefore, the selective security model sufficiently covers the vast majority of threat scenarios in real-world healthcare applications.

## 5. BMHADS scheme

Before presenting the specific details of the proposed BMHADS scheme, Table 1 summarizes the key symbols used in the scheme's development.

### 5.1. Blockchain-based DKG protocol

As shown in Fig 4, the proposed DKG protocol consists of seven phases, namely registration, initialization, commitment and share distribution, complaint, key reconstruction, signature, and key update. The detailed construction process of each phase is as follows.

**1. Registration**

This protocol allows $P$ assistant nodes to jointly generate a global public key $PK_{group}$ and the corresponding private key share in the presence of up to $f$ Byzantine nodes. The protocol is built on an asynchronous network model and satisfies the honest-majority condition $P \geq 3f + 1$, a prerequisite for achieving BFT. The agreement process is as follows:

Device self-registration: A medical terminal device $Dev_d$ $(d \in \{1, \ldots, D'_{total}\})$ independently generates a private key $skk_d = \Gamma_d \in \mathbb{Z}_r^*$ locally and calculates the corresponding identity public key $pk_d = g_2^{\Gamma_d}$. The device then uploads its public key, $pk_d$, and identity identifier, $ID_d$, to the blockchain smart contract for anchoring.

Node registration: Before the protocol initialization, all ANs intending to join the collaborative network must complete identity registration on the blockchain. Given the public and transparent nature of the blockchain ledger, each assistant node $AN_u$ must submit its unique identifier $ID_u$ and on-chain address $addr_u \in Addr$ to the smart contract, where $Addr = \{addr_1, addr_2, \ldots, addr_P\}$. Additionally, the smart contract incorporates an economic penalty mechanism for malicious

**Table 1. Notation description.**

| Notations | Description |
|---|---|
| $(pk_d, skk_d)$ | Key pair of the device |
| $(PK_{u,sign}, SK_{u,sign})$ | Signing key pair of the assistant node |
| $(PK_{u,enc}, SK_{u,enc})$ | Encryption key pair of the assistant node |
| $(PK_{group}, SK_{group})$ | Global group key pair |
| $\sigma_{group}$ | DKG protocol threshold signature |
| $V_{u,w}$ | Commitment coefficient |
| $s_{u,v}$ | Secret fragment |
| $sk_u$ | Local private key share |
| $\mathcal{H}$ | The set of honest assistant nodes |
| $SysPara$ | The public parameter set |
| $\tilde{\mathbb{T}}$ | Hierarchical access tree |
| $GID$ | Global identifier of users |
| $A_k$ | Attribute authority $k$ |
| $(PK_k, MSK_k)$ | Key pair of $A_k$ |
| $SK_U$ | Private key of data users |
| $M_m$ | Message |
| $\tilde{A}_k$ | Set of attributes managed by $A_k$ |
| $\tilde{A}_U$ | Set of attributes associated with DU |

nodes; therefore, assistant nodes must also deposit a security deposit of $Stake_u$ into the contract during registration. Upon completing registration, the node's status is activated, allowing it to participate formally in the DKG protocol. Furthermore, to ensure the security of subsequent distributed communications, each node $AN_u$ publishes an encryption key pair $(PK_{u,enc}, SK_{u,enc})$ and a signing key pair $(PK_{u,sign}, SK_{u,sign})$ to secure private share transmission and message authentication.

**2. Initialization**

First, define the set of assistant nodes as $\mathcal{AN} = \{AN_1, \ldots, AN_P\}$ and set the system security threshold to $t$. According to BFT requirements, the total number of nodes must satisfy $P \geq 3f + 1$, where $f$ is the maximum number of malicious nodes the system can tolerate. At this stage, each assistant node $AN_u$ independently selects a set of random coefficients $a_{u,w}$ locally and uses them to construct a random polynomial of order $t-1$ as shown below, where $u = \{1, \ldots, P\}$.

$$f_u(z) = a_{u,0} + a_{u,1}z + a_{u,2}z^2 + \ldots + a_{u,t-1}z^{t-1}$$

(2)

**3. Commitment and share distribution**

Each assistant node $AN_u$ generates a commitment $V_{u,w} = g_2 \cdot a_{u,w}$ for every coefficient of the polynomial, where $w \in \{0, \ldots, t-1\}$. These commitment coefficients are subsequently broadcast to all other assistant nodes to ensure data consistency across the network. After a successful distribution, the nodes submit the commitment set $\{V_{u,w}\}$ to the smart contract for on-chain anchoring.

To ensure the confidentiality and reliability of the source of the private key component $s_{u,v}$ during distribution, sender $AN_u$ performs the following operations: First, $AN_u$ computes the private key fragment $s_{u,v} = f_u(v) \mod r$, where $v = \{1, \ldots, P\}$, and encrypts component $s_{u,v}$ using the public key $PK_{v,enc}$ of recipient $AN_v$, resulting in $C_{u,v} = Enc(PK_{v,enc}, s_{u,v})$. Next, to prevent third parties from forging or tampering with the message, $AN_u$ uses its private signing key $SK_{u,sign}$ to sign the ciphertext, generating the proof $\sigma_{u,v} = Sig(SK_{u,sign}, C_{u,v})$. Finally, $AN_u$ sends the tuple $(C_{u,v}, \sigma_{u,v})$ to $AN_v$ via the peer-to-peer (P2P) network.

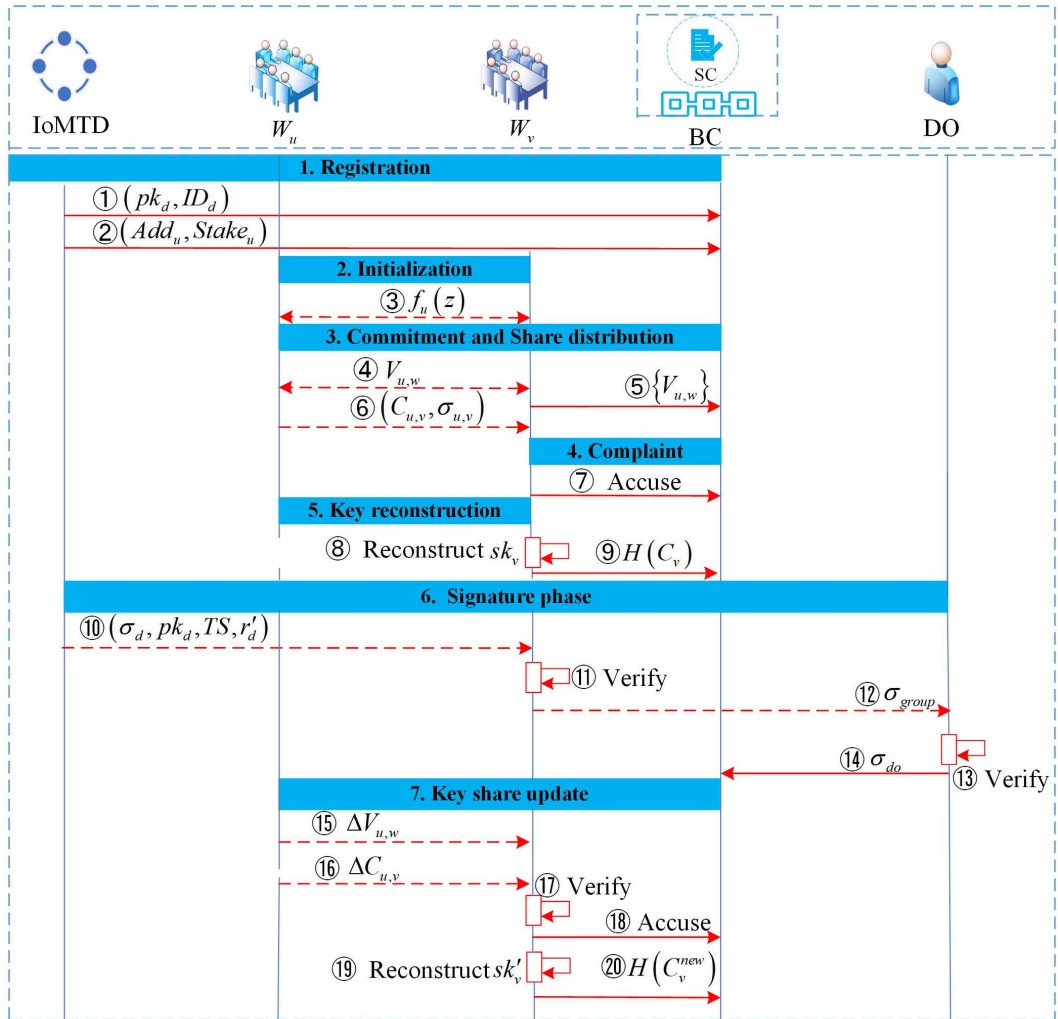

**Fig 4. DKG protocol process. Solid arrows indicate interactions with the blockchain, while dashed arrows indicate interactions between entities.** The step numbers correspond to the descriptions in the text.

To prevent malicious nodes from forging shards or executing replay attacks, upon receiving the data shard packet $(C_{u,v}, \sigma_{u,v})$ sent by sender $AN_u$, recipient $AN_v$ performs the following steps.

$AN_v$ first retrieves sender $AN_u$'s identity public key $PK_{u,sign}$ via a blockchain smart contract and executes verification logic on the signature $\sigma_{u,v}$. If the signature verification fails, the message is deemed untrustworthy and discarded.

Assuming the source is authentic, $AN_v$ decrypts $C_{u,v}$ using its locally held encryption private key $SK_{v,enc}$ to recover the secret share $s_{u,v}$.

To ensure that the distributor $AN_u$ has honestly executed the polynomial distribution protocol, $AN_v$ uses the decrypted $s_{u,v}$ in combination with the publicly available commitment $V_{u,w}$ on the smart contract to verify whether the following equation holds:

$$g_2 \cdot s_{u,v} = \sum_{w=0}^{t-1} V_{u,w} \cdot v^w \bmod r$$

(3)

## 4. Complaint

If the verification fails, the $AN_v$ will file a complaint against the $AN_u$ with the smart contract. The specific complaint process is as follows.

$AN_v$ does not simply discard the data. Instead, they immediately construct a publicly verifiable evidence package. This evidence package includes the on-chain address of the accused node $AN_u$, the private key share $s_{u,v}$, the corresponding ciphertext of the encrypted share $C_{u,v}$, the original signature $\sigma_{u,v}$, and the relevant coefficient commitment $V_{u,w}$, all published on-chain. Subsequently, $AN_v$ uses their local private key to sign this evidence package, obtaining $\sigma_{auth}$ digitally, and broadcasts it to other nodes.

Upon receiving the message, all honest nodes collectively execute the Byzantine Fault-Tolerant consensus protocol. First, they use $PK_{v,sign}$ to verify whether the signature $\sigma_{auth}$ validates the accusation initiated by $AN_v$; if so, they proceed to vote according to the protocol.

If it is confirmed that $AN_u$ sent an erroneous share, the contract will automatically deduct the collateral staked by $AN_u$, remove $AN_u$ from the list of qualified nodes, and invalidate $AN_u$'s registration information and public commitment. Then, the smart contract broadcasts the latest blocklist, $\mathcal{F}$. The remaining assistant nodes update their local set of qualified participants based on the consensus conclusion, remove the cheating node's weight, and renegotiate the system's public key.

## 5. Key reconstruction

After the validation and consensus phases are complete, each honest assistant node $AN_u$ will reconstruct the global public key and its local private key share. The global public key $PK_{group}$ is established by aggregating the initial commitments of all participants, calculated as follows:

$$PK_{group} = \sum_{u=1}^{P} V_{u,0} \bmod r$$

(4)

Each assistant node $AN_v$ obtains its local private key share $sk_v$ by aggregating the valid fragments received from all $P$ nodes.

$$sk_v = \sum_{u=1}^{P} s_{u,v} \bmod r$$

(5)

Subsequently, each honest assistant node $AN_v$ calculates its encrypted share $C_v = Enc\left(PK_{v,enc}, sk_v\right)$ and the corresponding hash value $H\left(C_v\right)$, and submits them to the blockchain as tamper-proof on-chain evidence. According to the protocol design, although the global master private key $SK_{group}$ is theoretically computable, the full key cannot be reconstructed unless the predefined collusion threshold $t$ is exceeded.

$$SK_{group} = \sum_{u=1}^{P} a_{u,0} \bmod r$$

(6)

## 6. Signature

After the DKG protocol completes the initialization of distributed trust anchors, the system enters the real-time data authentication phase. This phase aims to establish a dual-verification defense mechanism against unauthorized data injection through local pre-authentication on IoMT terminals and distributed threshold signatures on assistant nodes. The specific formalized process is as follows:

Each IoMT terminal device $Dev_d$ $\left(d \in \{1, \ldots, D'_{total}\}\right)$ collects raw medical data $M_d$. The device uses the private key $skk_d = \Gamma_d \in \mathbb{Z}_r^*$, which it generated autonomously during the registration phase, to execute the BLS lightweight signature algorithm. To ensure the uniqueness of the data packet and defend against replay attacks, the device constructs a data packet containing the current timestamp $TS$ and a random number $r'_d$, and calculates the signature value $\sigma_d$ as follows:

$$\sigma_d = \Gamma_d \cdot H\left(M_d \| TS \| r'_d \| ID_d\right)$$

(7)

Subsequently, the device broadcasts the data packet $\left(\sigma_d, pk_d, TS, r'_d\right)$ to its associated assistant node pool.

Upon receiving the packet $\left(\sigma_d, pk_d, TS, r'_d\right)$, the assistant node $AN_v$ first verifies the timeliness of the data by checking the timestamp offset $\left(TS_{current} - TS\right) < \Delta t$. Subsequently, the assistant node retrieves the device's anchored public key, $pk_d$, from the blockchain and verifies the legitimacy of the device's signature through a bilinear pairing operation, as shown below:

$$\hat{e}\left(g_2, \sigma_d\right) = \hat{e}\left(pk_d, H\left(M_d \| TS \| r'_d \| ID_d\right)\right)$$

(8)

If the equation holds, the process proceeds to the next step. Otherwise, the data is discarded, and an error log is recorded.

In high-frequency data transmission scenarios, such as real-time IoMT monitoring, this scheme introduces a batch verification mechanism to further reduce the computational overhead on assistant nodes. Suppose assistant node $AN_v$ receives $N_p$ consecutive data packets and their corresponding signatures $\sigma_{d,p}$ $\left(p \in \{1, \ldots, N_p\}\right)$ from the same device $Dev_d$, where the device's unique identifier is $ID_d$, the anchored public key is $pk_d$, and $p$ is the packet index sequence for the current batch task. The node uses the additive homomorphism of the bilinear mapping over $\mathbb{G}_1$ to accumulate the $N_p$ signature components and their corresponding message hash map values in the group space. The message hash map values are computed from the original data $M_{d,p}$, the timestamp $TS_p$, and the random number $r'_{d,p}$. The specific verification equation is shown in the following formula.

$$\hat{e}\left(\sum_{p=1}^{N_p} \sigma_{d,p}, g_2\right) = \hat{e}\left(\sum_{p=1}^{N_p} H(M_{d,p} \| TS_p \| r'_{d,p} \| ID_d), pk_d\right)$$

(9)

After verifying the authenticity of the IoMT terminal data source through single-point or batch verification, each assistant node $AN_v$ invokes the local private key share $sk_v$ obtained during the DKG phase to generate a local signature $\sigma'_v$ for the current medical data. Here, $h' = H\left(M_d \| TS \| tag\right)$.

$$\sigma'_v = sk_v \cdot h'$$

(10)

When any node receives at least $t$ valid signature fragments $\sigma'_v$, it uses the Lagrange interpolation coefficient $\gamma_v\left(z\right)$ to aggregate and generate a system-level threshold signature $\sigma_{group}$.

$$\sigma_{group} = \sum_{v=1}^{t} \gamma_v\left(z\right) \cdot \sigma'_v$$

(11)

After the DO collects the threshold signature $\sigma_{group}$, it verifies its validity using the system-wide public key $PK_{group}$ as follows.

$$\hat{e}\left(\sigma_{group}, g_2\right) = \hat{e}\left(h', PK_{group}\right)$$

(12)

After successful verification, DO generates a public-private key pair $(pk_{do}, sk_{do})$ in a secure environment, selects the current transaction timestamp $ts_1$, and concatenates the verified threshold signature $\sigma_{group}$, the original message hash $h'$, and the personal public key $pk_{do}$ to construct the final notarized digest $h'' = H(ts_1 \| \sigma_{group} \| h' \| pk_{do})$, which is then packaged and uploaded to the BC.

$$\sigma_{do} = sk_{do} \cdot h'' \tag{13}$$

### 7. Key update

To counter attacks from mobile adversaries and ensure the system's forward security, the proposed scheme introduces a share update mechanism based on active secret sharing. This mechanism allows assistant nodes to periodically evolve their private key shares without changing the global public key $PK_{group}$.

When the system enters the preset update cycle $T+1$, each eligible assistant node $AN_u$ initiates a local update algorithm. $AN_u$ randomly selects a $t-1$ time random polynomial $\Delta f_u(z)$, with its constant term set to 0, i.e., $\Delta f_u(0) = 0$. Simultaneously, $AN_u$ computes and broadcasts the increment commitment coefficient $\Delta V_{u,w}$ to other assistant nodes $AN_v$.

$$\Delta f_u(z) = 0 + b_{u,1}z + b_{u,2}z^2 + \ldots + b_{u,t-1}z^{t-1} \tag{14}$$

$AN_u$ calculates the share increment $\Delta s_{u,v} = \Delta f_u(v) \bmod r$ for other nodes $AN_v$, encrypts this increment $\Delta C_{u,v} = Enc(PK_{v,enc}, \Delta s_{u,v})$ using $AN_v$'s public key $PK_{v,enc}$, and sends it to $AN_v$.

Upon receiving $\Delta C_{u,v}$, $AN_v$ decrypts $\Delta C_{u,v}$ using their private key $SK_{v,enc}$ to obtain $\Delta s_{u,v}$, and performs verifiability checks using the public increment commitment as follows:

$$g_2 \cdot \Delta s_{u,v} = \sum_{w=1}^{t-1} \Delta V_{u,0} \cdot \Delta V_{u,w} \cdot v^w \bmod r \tag{15}$$

where $\Delta V_{u,0} = 1$. If the verification fails, the aforementioned challenge process is triggered.

After confirming that all received increment shards are valid, $AN_v$ computes the private key share $sk'_v$ for the new round.

$$sk'_v = sk_v + \sum_{u=1}^{P} \Delta s_{u,v} \pmod{r} \tag{16}$$

Once the above updates are complete, $AN_v$ recalculates the new local share ciphertext $C_v^{new} = Enc(PK_{v,enc}, sk'_v)$ and its hash value $H(C_v^{new})$, upload $H(C_v^{new})$ to the blockchain, and overwrite the old hash value $H(C_v)$.

## 5.2. Data sharing phase

Fig 5 depicts the complete interaction process for data sharing, with key steps as follows.

### 1. System Setup

$CA.Setup(1^\kappa) \to SysPara$. CA first takes the security parameter $\kappa$ as input and sets three groups $\mathbb{G}_1$, $\mathbb{G}_2$, and $\hat{\mathbb{G}}_T$, each of prime order $r$. A bilinear map $\hat{e}: \mathbb{G}_1 \times \mathbb{G}_2 \to \hat{\mathbb{G}}_T$ exists. Let $g_1$ be a generator of $\mathbb{G}_1$, and $g_2$, $h_2$ be generators of $\mathbb{G}_2$. Define four hash functions: $H_0, H_1: \{0,1\}^* \to \mathbb{G}_1$, $H_2: \{0,1\}^* \to \mathbb{Z}_r^*$, $H_3: \{0,1\}^* \to \hat{\mathbb{G}}_T$. For each global identity GID, compute $\phi = H_2(GID)$. The system public parameter is $SysPara = \{\hat{e}, r, g_1, g_2, h_2, \mathbb{G}_1, \mathbb{G}_2, \hat{\mathbb{G}}_T, H_0, H_1, H_2, H_3\}$.

### 2. Authority Setup

$Auth.Setup(SysPara) \to (MSK_k, PK_k)$. Each attribute authority $AA_k$ chooses a random exponent $\delta_k \in \mathbb{Z}_r^*$ as its master secret key component and computes the public key component $\psi_k = \hat{e}(g_1, g_2)^{\delta_k}$. For each attribute $\chi_{k,i} \in \tilde{A}_k$ under its

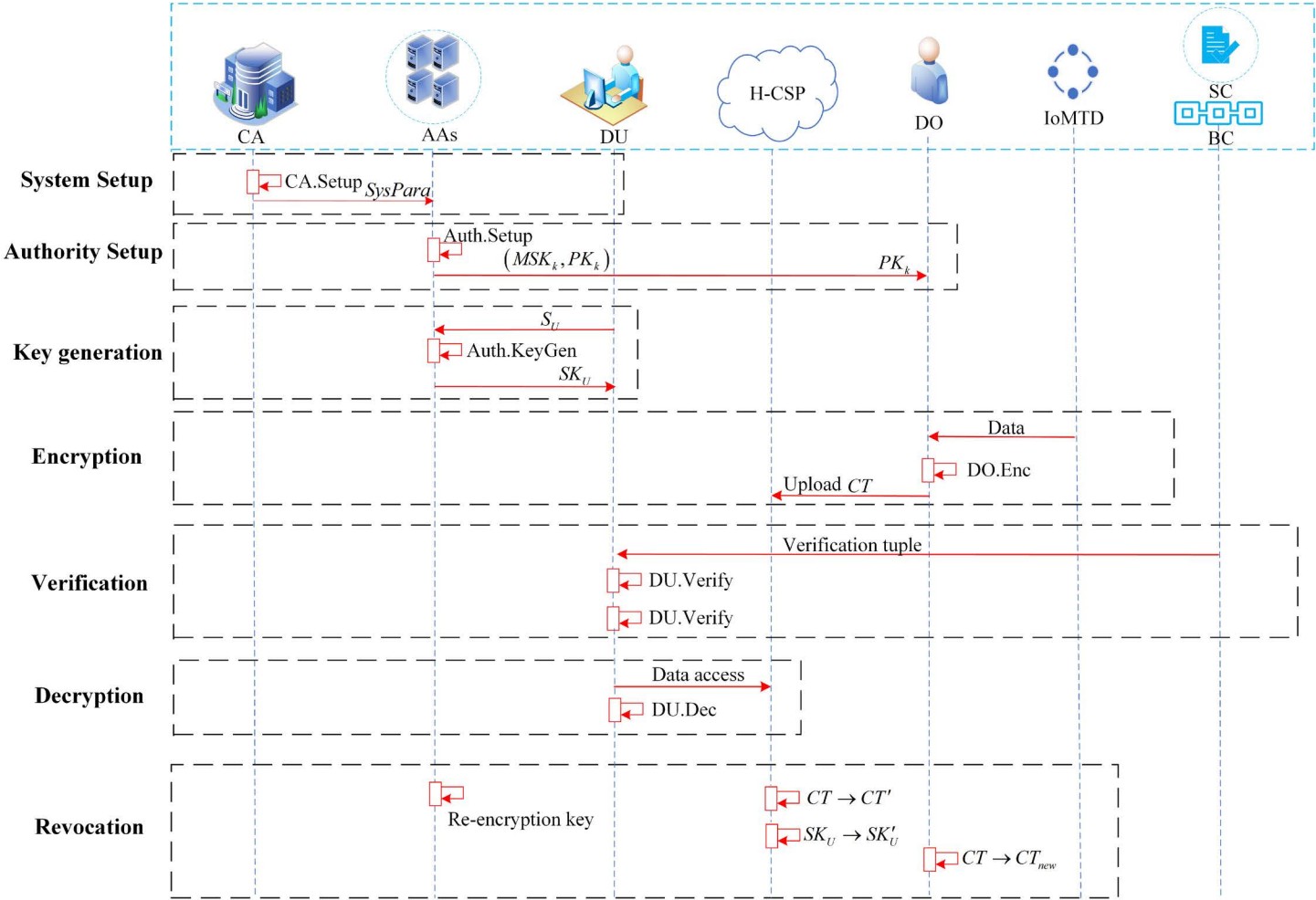

**Fig 5. Data sharing process.**

jurisdiction, a random value $\mathcal{K}_{k,i} \in \mathbb{Z}_r^*$ is selected to compute the corresponding public key component $\beta_{k,i} = g_1^{\mathcal{K}_{k,i}}$. To facilitate anonymous exchange, any two authorities $A_k$ and $A_j$ choose secret values $x_k, x_j \in \mathbb{Z}_r^*$ and establish a shared seed $\lambda_{kj} = \lambda_{jk}$ via a secured channel, which remains confidential between them. The pseudo-random function is thus defined as $PRF_{kj}(\phi) = h_2^{\frac{x_k x_j}{\lambda_{kj}+\phi}}$. Additionally, each authority initializes a cycle identifier $epoch = 1$, which is embedded into the keys. The master secret key ($MSK_k$ and the public key $PK_k$ are defined as follows:

$$MSK_k = \left\{epoch, \delta_k, x_k, \lambda_{kj}, \mathcal{K}_{k,i}\right\}_{j \in \{1,\dots,N\}\setminus\{k\}, i \in \{1,\dots,n_k\}} \tag{17}$$

$$PK_k = \left\{epoch, \psi_k, \beta_{k,i}\right\}_{i \in \{1,\dots,n_k\}} \tag{18}$$

### 3. Key generation

$Auth.KeyGen\left(MSK_k, GID, S_U\right) \rightarrow SK_U$. When a user requests a secret key using their global identifier $GID$, the attribute authority $AA_k$ computes $\phi = H_2(GID)$ and chooses a random value $\vartheta_k \in \mathbb{Z}_r^*$. For each user attribute $\chi_{k,i} \in \tilde{A}_U^k$ (where $\tilde{A}_U^k = \tilde{A}_U \cap \tilde{A}_k$), $AA_k$ calculates the attribute key component $S_{k,i} = h_2^{\vartheta_k / \mathcal{K}_{k,i}}$. Regarding the anonymous exchange protocol, the intermediate component $D_{kj}$ is computed based on the indices $k$ and $j$: If $k>j$, $D_{kj} = g_2^{\delta_k} \cdot h_2^{\vartheta_k} \cdot PRF_{kj}(\phi)$; If $k<j$, $D_{kj} = g_2^{\delta_k} \cdot h_2^{\vartheta_k} / PRF_{kj}(\phi)$. Finally, the user aggregates these components to obtain the complete secret key $SK_U = \left\{epoch, D_U, S_{k,i}\right\}_{\chi_{k,i} \in \tilde{A}_U^k, k \in \{1, \dots, N\}}$, where $D_U$ is defined in the following formula.

$$D_U = \prod_{(k,j) \in \{1, \dots, N\} \times (\{1, \dots, N\} \setminus \{k\})} D_{kj}$$
$$= g_2^{\sum_{k \in \{1, \dots, N\}} (N-1)\delta_k} \cdot h_2^{\sum_{\{k \in 1, \dots, N\}} (N-1)\vartheta_k} \tag{19}$$

### 4. Encryption

$(4)DO.Enc\left(PK_k, M, \tilde{\mathbb{T}}\right) \rightarrow CT$. Given a dataset $M = \{M_1, \dots, M_l\}$ with $l$ access levels, DO constructs a hierarchical access tree $\tilde{\mathbb{T}}$ and performs the following encryption:

First, DO selects a random content key $ck_m$ for each level $m \in [1, l]$ and computes the level ciphertext components $\hat{C}_m = Enc_{ck_m}(M)$. Next, DO selects a polynomial $q_{(x,y)}$ of degree $th_{(x,y)} - 1$ for each node $(x,y)$ in $\tilde{\mathbb{T}}$. For nodes $(x_m, y_m)$ representing a level, set the value of their polynomial at the origin to satisfy $q_{(x_m, y_m)}(0) = \rho_m$, where $\rho_m \in \mathbb{Z}_r^*$ is the random secret value for each level. For other nodes $(x,y)$ in the access tree, the value of their polynomial is determined by the parent node, i.e., $q_{(x,y)}(0) = q_{parent}(index(x, y))$. Thus, DO uses the level secret $\rho_m$ to construct the CP-ABE ciphertext components $\tilde{C}_m$ and $C_m'$ corresponding to the level message as follows:

$$\begin{cases} \tilde{C}_m = ck_m \cdot \left(\prod_{k \in \{1, \dots, N\}} \psi_k\right)^{\rho_m} \\ \\ C_m' = g_1^{\rho_m} \end{cases} \tag{20}$$

After establishing the hierarchical tree structure of the hierarchical access tree $\tilde{\mathbb{T}}$, DO further constructs fine-grained ciphertext components for different types of nodes in $\tilde{\mathbb{T}}$. Specifically, for each leaf node $(x, y)$, the associated attribute value is set to $\chi_{k,i} = att(x, y)$. DO performs a blinding operation using the value $q_{(x, y)}(0)$ of the node's polynomial at the origin, thereby constructing the attribute-related ciphertext components $C_{k,(x,y)}$ and $\breve{C}_{(x,y)}$. Furthermore, if node $(x, y)$ is a transport node, the set of its subordinate sub-threshold nodes is defined as $TN-SET(x, y) = \{ch_1, \dots, ch_{j'}\}$. Based on this, DO further computes and generates the transport node ciphertext $\bar{C}_{(x,y),j'}$. The detailed construction equations for the relevant attribute ciphertext components and transport node components are shown below:

$$\begin{cases} C_{k,(x,y)} = \left\{\beta_{k,i}^{q_{(x,y)}(0)}\right\}_{\chi_{k,i} \in \tilde{\mathbb{T}}} \\ \breve{C}_{(x,y)} = H_1\left(att(x, y)\right)^{q_{(x,y)}(0)} \\ \bar{C}_{(x,y),j'} = \left(\prod_{k \in \{1, \dots, N\}} \psi_k\right)^{q_{(x,y)}(0) + q_{ch_{j'}}(0)} \cdot H_3\left(\prod_{k \in \{1, \dots, N\}} \psi_k\right)^{q_{(x,y)}(0)} \end{cases} \tag{21}$$

Finally, DO outputs the ciphertext $CT = \left\{\tilde{\mathbb{T}}, \tilde{C}_m, C_m', C_{k,(x,y)}, \breve{C}_{(x,y)}, \bar{C}_{(x,y),j'}\right\}$.

### 5. Verification

$DU.Verify\left(SysPara, \Lambda, PK_{group}, pk_{do}\right) \rightarrow \{0, 1\}$. After successfully downloading the ciphertext, the data user must first retrieve and verify the tuple $\Lambda$ from the blockchain. The overall verification process consists of the following two core stages. Specifically, the tuple is defined as $\Lambda = \left\{\sigma_{group}, \sigma_{do}, h', h'', ts_1\right\}$.

The first stage verifies the validity of the threshold signature. This step ensures that the participating working nodes have reached mathematical consensus and that the data source is trustworthy. The specific verification equation is as follows:

$$\hat{e}\left(\sigma_{group}, g_2\right) = \hat{e}\left(h', PK_{group}\right) \tag{22}$$

The second stage involves verifying the secondary signature executed by the data owner. This operation further enhances the data's non-repudiation, and its verification equation is as follows:

$$\hat{e}\left(\sigma_{do}, g_2\right) = \hat{e}\left(h'', pk_{do}\right) \tag{23}$$

If both of the above verification equations hold, the algorithm outputs 1 and formally initiates the hierarchical CP-ABE decryption process. If either equation fails verification, the data user will deem the data packet untrustworthy and immediately terminate the current operation. To demonstrate the rigor and completeness of the above verification logic, this section provides the relevant correctness proof below.

**Correctness.** The validity of the threshold signature verification is demonstrated through the following derivation.

$$\hat{e}\left(\sigma_{group}, g_2\right) = \hat{e}\left(h', PK_{group}\right)$$
$$\hat{e}\left(\sum_{v \in H} \gamma_v(z)\left(sk_v \cdot h'\right), g_2\right) = \hat{e}\left(h', PK_{group}\right)$$
$$\hat{e}\left(h', g_2\right)^{\sum_{v \in \mathcal{H}} \gamma_v(z) \cdot sk_v} = \hat{e}\left(h', g_2^{a_{v,0}}\right)$$
$$\hat{e}\left(h', g_2\right)^{a_{v,0}} = \hat{e}\left(h', g_2\right)^{a_{v,0}} \tag{24}$$

Similarly, the correctness of the secondary signature generated by the DO is verified as follows:

$$\hat{e}(\sigma_{do}, g_2) = \hat{e}(h'', pk_{do})$$
$$\hat{e}(sk_{do} \cdot h'', g_2) = \hat{e}(h'', g_2^{sk_{do}})$$
$$\hat{e}(h'', g_2)^{sk_{do}} = \hat{e}(h'', g_2)^{sk_{do}} \tag{25}$$

### 6. Decryption

$DU.Dec\left(PK_k, CT, SK_U\right) \rightarrow CT$. DU executes this algorithm to recover the message content key $ck_m$ at a specific level using the set of multi-authority attribute private keys it holds. The core of the decryption process lies in processing the hierarchical access tree $\tilde{\mathbb{T}}$ in the ciphertext by calling the recursive algorithm $DecNode(CT, SK_U, (x, y))$. The detailed execution logic of this algorithm is as follows:

If node $(x, y)$ is a leaf node of the hierarchical access tree, the execution logic of the recursive function $DecNode(CT, SK_U, (x, y))$ is as follows: If its associated attribute is $\chi_{k,i} \notin \tilde{A}_k^U$, then since the attribute does not satisfy the access policy, $DecNode(CT, SK_U, (x, y)) = \perp$ is returned. Conversely, if its associated attribute is $\chi_{k,i} \in \tilde{A}_k^U$, the computation process for $DecNode(CT, SK_U, (x, y))$ is as shown below:

$$DecNode(CT, SK_U, (x, y)) = \prod_{k \in \{1, ..., N\}} \hat{e}\left(C_{k,(x,y)}, S_{k,i}\right)$$

$$= \prod_{k \in \{1, ..., N\}} \hat{e}\left((\beta_{k,i})^{q_{(x,y)}(0)}, h_2^{\frac{\vartheta_k}{\mathcal{K}_{k,i}}}\right)$$

$$= \prod_{k \in \{1, ..., N\}} \hat{e}(g_1^{\mathcal{K}_{k,i} \cdot q_{(x,y)}(0)}, h_2^{\frac{\vartheta_k}{\mathcal{K}_{k,i}}})$$

$$= \hat{e}(g_1, h_2)^{q_{(x,y)}(0) \cdot (\sum_{k \in \{1, ..., N\}} \vartheta_k)} \tag{26}$$

If node $(x, y)$ is a non-leaf node, the algorithm employs a bottom-up recursive reconstruction strategy. First, for each child node $Z$ of node $(x, y)$, $F_Z = DecNode(CT, SK_U, Z)$ is computed recursively, and $S_{(x,y)}$ is defined as the set of all child nodes for which the return result is not empty. The algorithm then performs a threshold check: if the cardinality of the set $|S_{(x,y)}|$ is less than the threshold value $th_{(x,y)}$, it indicates that the subtree branch cannot satisfy the access policy, and the function returns the empty value $\perp$. Conversely, if the condition $|S_{(x,y)}| \geq th_{(x,y)}$ is satisfied, select any $th_{(x,y)}$ child nodes from S to form a subset $S'_{(x,y)}$, where $S'_{(x,y)} = \{i = index(Z), Z \in S_{(x,y)}\}$. The specific reconstruction equation is as follows:

$$F_{(x,y)} = \prod_{Z \in S_{(x,y)}} F_Z^{\Delta_{i,S'_{(x,y)}}(0)}$$

$$= \prod_{Z \in S_{(x,y)}} \left(\hat{e}(g_1, h_2)^{q_Z(0) \cdot \left(\sum_{k \in \{1, ..., N\}} \vartheta_k\right)}\right)^{\Delta_{i,S'_{(x,y)}}(0)}$$

$$= \prod_{Z \in S_{(x,y)}} \left(\hat{e}(g_1, h_2)^{q_{(x,y)}(i) \cdot \left(\sum_{k \in \{1, ..., N\}} \vartheta_k\right)}\right)^{\Delta_{i,S'_{(x,y)}}(0)}$$

$$= \hat{e}(g_1, h_2)^{q_{(x,y)}(0) \cdot \left(\sum_{k \in \{1, ..., N\}} \vartheta_k\right)} \tag{27}$$

Thereafter, the decryption algorithm is executed to retrieve the content key $ck_m$. Provided that the attribute set of the DU satisfies the hierarchical access tree $\tilde{\mathbb{T}}$ either partially or fully, the intermediate components $\hat{e}(g_1, g_2)^{\rho_m \vartheta_k}$ are derived via the recursive operations described previously, where $m \in \{1, ..., l\}$ and $k \in \{1, ..., N\}$. Specifically, for each level node $(x_m, y_m)$, the target bilinear pairing value $\hat{e}(g_1, g_2)^{\rho_m \sum_{k \in \{1, ..., N\}} \delta_k}$ is reconstructed as follows:

$$F_m = \frac{\hat{e}(C'_m, D_U)^{\frac{1}{N-1}}}{DecNode(CT, SK_U, (x_m, y_m))}$$

$$= \frac{\hat{e}\left(g_1^{\rho_m}, g_2^{\sum_{k \in \{1,2,...,N\}}(N-1)\delta_k} \cdot h_2^{\sum_{k \in \{1,2,...,N\}}(N-1)\vartheta_k}\right)^{\frac{1}{N-1}}}{\hat{e}(g_1, h_2)^{\rho_m \cdot \left(\sum_{k \in \{1,2,...,N\}} \vartheta_k\right)}}$$

$$= \hat{e}(g_1, g_2)^{\rho_m \cdot \left(\sum_{k \in \{1,2,...,N\}} \delta_k\right)} \tag{28}$$

Assuming $\tilde{A}_k^U$ satisfies all underlying nodes, the ciphertext element $\bar{C}_{(x,y),j'}$ enables the recursive reconstruction of hierarchical values. Consequently, the sequence $\{F_{(m+1),j'}, ..., F_{(l),j'}\}$ is retrieved in succession.

$$F_{(m+1),j'} = \frac{\bar{C}_{(x_m,y_m),j'}}{H_3(F_m) \cdot F_m}$$

$$= \frac{\left(\prod_{k \in \{1,\ldots,N\}} \psi_k\right)^{q_{(x_m,y_m)}(0)+q_{ch_{j'}}(0)} \cdot H_3\left(\prod_{k \in \{1,\ldots,N\}} \psi_k\right)^{q_{(x_m,y_m)}(0)}}{H_3\left(\prod_{k \in \{1,\ldots,N\}} \psi_k\right)^{\rho_m} \cdot \left(\prod_{k \in \{1,\ldots,N\}} \psi_k\right)^{\rho_m}}$$

$$= \left(\prod_{k \in \{1,\ldots,N\}} \hat{e}(g_1, g_2)^{\delta_k}\right)^{q_{ch_{j'}}(0)}$$

$$= \hat{e}(g_1, g_2)^{q_{ch_{j'}}(0) \cdot \sum_{k \in \{1,\ldots,N\}} \delta_k} \tag{29}$$

As demonstrated in the derivation below, for each hierarchy level $m \in [1, l]$ where the access policy is satisfied, the DU can precisely recover the corresponding level content key $ck_m$ through algebraic cancellation:

$$ck_m = \frac{\tilde{C}_m}{F_m}$$

$$= \frac{ck_m \cdot \hat{e}(g_1, g_2)^{\rho_m \sum_{k \in \{1,\ldots,N\}} \delta_k}}{\hat{e}(g_1, g_2)^{\rho_m \sum_{k \in \{1,\ldots,N\}} \delta_k}}$$

$$= ck_m \ (m \in [1, l]) \tag{30}$$

Upon acquiring the hierarchical key $ck_m$, the user invokes the corresponding symmetric decryption algorithm $Dec_{ck_m}(\hat{C}_m)$ to decrypt the hierarchical ciphertext $\hat{C}_m$, thereby recovering the plaintext message $M_m$ for each level $m \in [1, l]$.

### 7. Revocation

When revoking a specific attribute set $R'$, the relevant attribute authority $A_k$ performs this operation for each attribute $\chi_{k,i} \in R'$ to be updated. The authority selects a new parameter $\mathcal{K}'_{k,i} \in \mathbb{Z}_r^*$, calculates a new public key component $\beta'_{k,i} = g_1^{\mathcal{K}'_{k,i}}$, and re-encrypts the key $\Delta_{i \to i'(n)} = \mathcal{K}_{k,i}(n) \cdot \mathcal{K}_{k,i}^{-1}$. Subsequently, the authority transitions the system into a new security *epoch* and outputs the local re-encryption key $\Delta_k$. The $\Delta_k$ values from all authorities collectively form the global re-encryption key $\Delta$.

The medical cloud server executes the algorithm. First, it verifies the *epoch* identifier of the ciphertext $CT$ against the re-encryption key $\Delta$. If the version is not the latest, the server calculates a new ciphertext component $C''_{k,(x,y)} = (C_{k,(x,y)})^{\Delta_{i \to i'(n)}} = g_1^{\vartheta_k \cdot \mathcal{K}_{k,i}}$ for all attributes $\chi_{k,i}$ in the hierarchical access structure $\tilde{\mathbb{T}}$ using the re-encryption key chain $\Delta_{i \to i'(n)}$. Finally, the server outputs the new ciphertext $CT' = \left\{ epoch', \tilde{C}_m, C'_m, C''_{k,(x,y)}, \check{C}_{(x,y)}, \bar{C}_{(x,y),j'} \right\}$ with the epoch identifier refreshed.

For non-revoked users, their keys need to be updated to decrypt new-cycle ciphertexts. Like ciphertext re-encryption, the server checks the epoch of the user key $SK_U$. For each attribute $\chi_{k,i}$ owned by the user, the server computes the updated key component $S'_{k,i} = (S_{k,i})^{\frac{1}{\Delta_{i \to i'(n)}}} = h_2^{\frac{1}{\mathcal{K}_{k,i}(n)}}$. Finally, it outputs the new key $SK'_U = \left\{ epoch', D_U, S'_{k,i} \right\}_{k \in \{1,\ldots,N\}, \chi_{k,i} \in \tilde{A}_k^U}$ with synchronized epoch updates.

When policies update, DO does not need to download and re-encrypt entire files. The algorithm only regenerates the ciphertext component $\hat{C}_{k,(x,y)}$ for attributes that change the access structure. Specifically, for newly added or modified attribute nodes $(x, y)$, DO computes $\hat{C}_{k,(x,y)}$ based on their new polynomial share value $q_{(x,y)}(0)$. Finally, the algorithm outputs an updated ciphertext $CT_{new} = \left\{ epoch, \tilde{C}_m, C'_m, \hat{C}_{k,(x,y)}, \check{C}_{(x,y)}, \bar{C}_{(x,y),j'} \right\}$ containing most of the original components alongside the newly generated component.

## 6. Security analysis

This section describes the security analysis of the proposed BMHADS scheme.

**Theorem 1 (IND-CPA security).** Assuming that the DBDH assumption holds, it is computationally infeasible for any probabilistic polynomial-time adversary to break the proposed BMHADS scheme under chosen-plaintext attacks. Therefore, the BMHADS scheme effectively achieves IND-CPA security.

Assume an adversary $\mathcal{A}$ with polynomial-time computational capability can exploit the benefits of $\varepsilon$ to compromise this system. Subsequently, we construct a simulator $\mathcal{B}$ to refute the hypothesis, which will break the DBDH hypothesis with $\varepsilon' \geq \frac{\varepsilon}{2} \cdot \prod_{k \in \{1,\ldots,N\}} \left(1 - \frac{n_k - 2}{(r-1)^2}\right)$, Where $n_k$ indicates the total count of attributes under the control of authority $A_k$.

**Proof:** $\mathcal{C}$ selects three cyclic groups $\mathbb{G}_1$, $\mathbb{G}_2$, $\hat{\mathbb{G}}_T$ of prime order $r$, along with corresponding generators $g_1 \in \mathbb{G}_1$ and $g_2, h_2 \in \mathbb{G}_2$. It randomly chooses $a, b, c \in \mathbb{Z}_r^*$ and then uniformly samples a random bit $\mu \in \{0, 1\}$:

If $\mu = 0$, $\mathcal{C}$ sends the real tuple $\left(g_1^a, g_1^b, g_1^c, g_2^a, g_2^b, g_2^c, \hat{e}(g_1, g_2)^{abc}\right)$ to $\mathcal{B}$.

If $\mu = 1$, $\mathcal{C}$ sends the random tuple $\left(g_1^a, g_1^b, g_1^c, g_2^a, g_2^b, g_2^c, \mathcal{Z}\right)$ to $\mathcal{B}$, where $\mathcal{Z}$ is a uniformly random element of $\hat{\mathbb{G}}_T$.

**System Setup:** At the start of the experiment, $\mathcal{A}$ selects a challenging hierarchical access tree structure $\tilde{\mathbb{T}}^*$ to be attacked and sends a set $\mathcal{V}_A$ of corrupt authorities under its control to simulator $\mathcal{B}$, where $\mathcal{V}_A$ satisfies the collusion constraint $\mathcal{V}_A \leq |N - 2|$. Subsequently, $\mathcal{B}$ randomly selects $\varpi \in \mathbb{Z}_r^*$ and computes $h_2 = g_2^{a+\varpi}$. Through this construction, $\mathcal{B}$ implicitly embeds the DBDH challenge a into the system parameter $h_2$. Due to the randomness of $\varpi$, the generated $h_2$ is statistically indistinguishable from the true parameter, thereby ensuring the soundness of the secure reduction.

**Authority Setup:** $\mathcal{B}$ randomly selects an authority $A_k$ from the set of honest authorities $A_k^* \in \{A_1, A_2, \ldots, A_N\} \setminus \mathcal{V}_A$ as the target authority for the challenge and implicitly embeds the DBDH puzzle instance into that authority's parameter configuration.

**Case 1:** For corrupted authorities $A_k \in \mathcal{V}_A$. Since the simulator $\mathcal{B}$ possesses the complete secret parameters of all corrupted authorities, it randomly selects $v_k, \omega_{k,i} \in \mathbb{Z}_r^*$ and computes $\tilde{\beta}_{k,i} = g_1^{\omega_{k,i}}$ according to the scheme's algorithm. For each attribute $\chi_{k,i} \in \tilde{A}_k$, $\mathcal{B}$ randomly chooses $\tilde{x}_k \in \mathbb{Z}_r^*$ and $\lambda_{kj}' \in \mathbb{Z}_r^*$. In this context, the PRF seed $\lambda_{kj}'$ is shared between two corrupted authorities $A_k$ and $A_j$. Subsequently, $\mathcal{B}$ transmits $\left\{v_k, \tilde{x}_k, \lambda_{kj}', \omega_{k,i}\right\}$ and $\left\{\tilde{\beta}_{k,i}, \psi_k'\right\}$ to the adversary $\mathcal{A}$, where $\psi_k' = \hat{e}(g_1, g_2)^{v_k}$.

**Case 2:** For honest authorities $A_k \in \mathcal{V}_A$. $\mathcal{B}$ selects $v_k, \omega_{k,i} \in \mathbb{Z}_r^*$. Regarding the hierarchical access tree $\tilde{\mathbb{T}}^*$, if attribute $\chi_{k,i} \in \tilde{\mathbb{T}}^*$, $\mathcal{B}$ computes $\tilde{\beta}_{k,i} = g_1^{\omega_{k,i}}$. Otherwise, if $\chi_{k,i} \notin \tilde{\mathbb{T}}^*$, $\mathcal{B}$ utilizes the component $g_1^b$ from the DBDH instance to compute $\tilde{\beta}_{k,i} = g_1^{b\omega_{k,i}}$. For the component $\psi_k'$, if $A_k \neq A_k^*$, $\mathcal{B}$ computes $\psi_k' = \hat{e}(g_1, g_2)^{bv_k}$. If $A_k = A_k^*$, $\mathcal{B}$ embeds the challenge term $ab$ into the public parameters by constructing $\psi_k'$ as follows:

$$\psi_k' = \hat{e}(g_1, g_2)^{ab} \cdot \prod_{A_k \in \mathcal{V}_A} \hat{e}(g_1, g_2)^{-v_k} \cdot \prod_{A_k \notin \mathcal{V}_A, A_k \neq A_k^*} \hat{e}(g_1, g_2)^{-bv_k}$$

(31)

Finally, $\mathcal{B}$ selects random seeds $\lambda_{kj}' \in \mathbb{Z}_r^*$ for honest authorities $A_k$ and $A_j$ to simulate the shared secrets, ensuring the transparency of PRF seed interactions to the adversary $\mathcal{A}$. Subsequently, $\mathcal{B}$ transmits the public components $(\tilde{\beta}_{k,i}, \psi_k')$, which incorporate the embedded DBDH challenge terms, to $\mathcal{A}$. This procedure achieves a statistically perfect simulation of the public parameter distribution in the real system.

**QueryPhase 1:** Under this security model, $\mathcal{A}$ can issue a sequence of private key queries for attribute sets $\{S_1, \ldots, S_{o+1}\}$. According to the security game definition, none of the queried attribute sets satisfy the challenge access structure $\tilde{\mathbb{T}}^*$ pre-selected by $\mathcal{A}$.

**Case 1:** For corrupted authorities $A_k \in \mathcal{V}_A$. Since $\mathcal{B}$ has already disclosed the complete secret parameters $\{v_k, \tilde{x}_k, \lambda_{kj}', \omega_{k,i}\}$ of these authorities to $\mathcal{A}$ during the authority setup phase, $\mathcal{B}$ directly invokes the actual algorithms defined in the scheme to generate the corresponding attribute secret keys.

**Case 2:** For honest authorities $A_k \in \mathcal{V}_A$. $\mathcal{B}$ constructs the key component $S_{k,i}$ depending on whether the attribute is involved in the challenge access structure:

If attribute $\chi_{k,i} \notin \tilde{\mathbb{T}}^*$, $\mathcal{B}$ randomly chooses $\vartheta_k \in \mathbb{Z}_r^*$ and computes $S_{k,i} = h_2^{\vartheta_k/(a+\varpi)\omega_{k,i}}$.

If attribute $\chi_{k,i} \in \tilde{\mathbb{T}}^*$, $\mathcal{B}$ computes $S_{k,i} = h_2^{\vartheta_k/\omega_{k,i}}$.

Regarding the interactive key component $D_{kj}$, $\mathcal{B}$ employs an algebraic cancellation technique to simulate the key structure containing the unknown term $g_2^{ab}$, categorized into two scenarios:

When authority $A_k$ is not the target authority ($A_k \neq A_k^*$), $\mathcal{B}$ possesses the auxiliary secret $bv_k$ and constructs:

If $k > j$, $D_{kj} = (g_2^b)^{\upsilon_k} \cdot h_2^{\vartheta_k} \cdot PRF_{kj}(\phi')$.

If $k < j$, $D_{kj} = (g_2^b)^{\upsilon_k} \cdot h_2^{\vartheta_k}/PRF_{kj}(\phi')$.

When authority $A_k$ is the target authority ($A_k = A_k^*$), the simulator $\mathcal{B}$ is unaware of the product $ab$ from the DBDH instance. To construct a valid key without this knowledge, $\mathcal{B}$ utilizes $\vartheta_k \in \mathbb{Z}_r^*$ as follows:

If $k > j$, $D_{kj} = (g_2^b)^{-\varpi} \cdot \prod_{A_k \in \mathcal{V}_A} g_2^{-\upsilon_k} \cdot \prod_{A_k \notin \mathcal{V}_A, A_k \neq A_k^*} (g_2^b)^{-\upsilon_k} \cdot h_2^{\vartheta_k} \cdot PRF_{kj}(\phi')$.

If $k < j$, $D_{kj} = \frac{(g_2^b)^{-\varpi} \cdot \prod_{A_k \in \mathcal{V}_A} g_2^{-\upsilon_k} \cdot \prod_{A_k \notin \mathcal{V}_A, A_k \neq A_k^*} (g_2^b)^{-\upsilon_k} \cdot h_2^{\vartheta_k}}{PRF_{kj}(\phi')}$.

Based on the analysis above, the distribution of $D_{kj}$ is statistically correct. Due to the mirror symmetry between $k > j$ and $k < j$, we focus on $k > j$ for brevity. In the following, we prove that $D_{kj}$ is mathematically consistent with the real scheme through algebraic equivalence derivation.

To demonstrate that the interactive key $D_{kj}$ constructed by the simulator $\mathcal{B}$ is mathematically consistent with the real scheme, we first provide its algebraic equivalence derivation. In this derivation, $\mathcal{B}$ expands the terms by directly utilizing the random exponent $\vartheta_k$ selected during the simulation process along with other known parameters:

$$
\begin{aligned}
D_{kj} &= (g_2^b)^{-\varpi} \cdot \prod_{A_k \in \mathcal{V}_A} g_2^{-\upsilon_k} \cdot \prod_{A_k \notin \mathcal{V}_A, A_k \neq A_k^*} (g_2^b)^{-\upsilon_k} \cdot h_2^{\vartheta_k} \cdot PRF_{kj}(\phi') \\
&= g_2^{-b\varpi} \cdot (g_2^{a+\varpi})^{\vartheta_k} \cdot g_2^{-\left(\sum_{A_k \in \mathcal{V}_A} \upsilon_k + \sum_{A_k \notin \mathcal{V}_A, A_k \neq A_k^*} bv_k\right)} \cdot PRF_{kj}(\phi') \\
&= (g_2^{a+\varpi})^{-b} \cdot g_2^{ab} \cdot (g_2^{a+\varpi})^{\vartheta_k} \cdot g_2^{-\left(\sum_{A_k \in \mathcal{V}_A} v_k + \sum_{A_k \notin \mathcal{V}_A, A_k \neq A_k^*} bv_k\right)} \cdot PRF_{kj}(\phi') \\
&= g_2^{ab} \cdot (g_2^a \cdot g_2^\varpi)^{\vartheta_k - b} \cdot g_2^{-\left(\sum_{A_k \in \mathcal{V}_A} \upsilon_k + \sum_{A_k \notin \mathcal{V}_A, A_k \neq A_k^*} bv_k\right)} \cdot PRF_{kj}(\phi') \\
&= g_2^{ab - \left(\sum_{A_k \in \mathcal{V}_A} \upsilon_k + \sum_{A_k \notin \mathcal{V}_A, A_k \neq A_k^*} bv_k\right)} \cdot h_2^{\vartheta_k - b} \cdot PRF_{kj}(\phi')
\end{aligned}
\tag{32}
$$

As illustrated in the derivation above, the core logic relies on expanding and consolidating exponential terms to achieve algebraic substitution. Specifically, $\mathcal{B}$ achieves algebraic cancellation by leveraging the pre-configured cancellation term $(g_2^b)^{-\varpi}$ and the public parameter $h_2$ to neutralize the term $g_2^{b\varpi}$ derived from the mapping involving the challenge component $b$. This step ensures that $\mathcal{B}$ can successfully retain the challenge term $g_2^{ab}$ (representing the master secret key) within the final key structure without requiring explicit knowledge of the secret $ab$ from the difficulty problem instance. By defining an implicit random mask $\vartheta_k' = \vartheta_k - b$, the interactive key $D_{kj}$ finally satisfies the following equation:

$$
D_{kj} = g_2^{ab - \left(\sum_{A_k \in \mathcal{V}_A} \upsilon_k + \sum_{A_k \notin \mathcal{V}_A, A_k \neq A_k^*} bv_k\right)} \cdot h_2^{\vartheta_k'} \cdot PRF_{kj}(\phi')
\tag{33}
$$

**Challenge:** $\mathcal{A}$ submits two equal-length messages $M_0$ and $M_1$ to the simulator $\mathcal{B}$. In response, $\mathcal{B}$ tosses a random coin $\xi \in \{0, 1\}$ and implicitly sets the secret value $s = c$. Utilizing the components from the DBDH instance, $\mathcal{B}$

constructs the challenge ciphertext $CT^* = \{\tilde{C}^*, C^*, C^*_{(x,y),k}, \bar{C}^*_{(x,y),j'}\}$, where $\tilde{C}^* = M_\xi \cdot \mathcal{Z}$, $C^* = g_1^c$, $C^*_{(x,y),k} = (\tilde{\beta}_{k,i})^{q_{(x,y)}(0)}$, and $\bar{C}^*_{(x,y),j'} = \mathcal{Z}^{q_{(x,y)}(0)+q_{ch_{j'}}(0)}$. The aforementioned construction establishes a rigorous connection between the scheme's security and the DBDH hardness assumption. If $\mu = 0$, it follows that $\mathcal{Z} = \hat{e}(g_1, g_2)^{abc}$. At this point, since $c$ is uniformly and randomly distributed in $\mathbb{Z}_r^*$, the simulated challenge ciphertext $CT^*$ is statistically indistinguishable from a valid ciphertext produced by the real encryption algorithm. Furthermore, by aggregating the public components of all authorities, the structural consistency of the simulated challenge ciphertext for $\mu = 0$ can be proved via the following derivation:

$$
\begin{aligned}
\prod_{k \in \{1,\dots,N\}} \psi_k^{'c} &= \prod_{A_k \in \mathcal{V}_A} \hat{e}(g_1, g_2)^{cv_k} \cdot \prod_{A_k \notin \mathcal{V}_A, A_k \neq A_k^*} \hat{e}(g_1, g_2)^{cbv_k} \cdot \hat{e}(g_1, g_2)^{abc} \\
&\quad \cdot \left( \prod_{A_k \in \mathcal{V}_A} \hat{e}(g_1, g_2)^{-cv_k} \prod_{A_k \notin \mathcal{V}_A, A_k \neq A_k^*} \hat{e}(g_1, g_2)^{-cbv_k} \right) \\
&= \hat{e}(g_1, g_2)^{abc} \\
&= \mathcal{Z}
\end{aligned}
\tag{34}
$$

When $\mu = 0$, algebraic cancellation of the public components from each authority causes the decryption term to coincide exactly with the challenge term $\mathcal{Z}$. Hence, $\mathcal{B}$ can construct a challenge ciphertext $CT^*$ that is statistically indistinguishable from a real-system ciphertext, thereby ensuring the rigor of the security reduction.

**QueryPhase 2:** This phase proceeds identically to QueryPhase 1.

**Guess:** The adversary $\mathcal{A}$ submits its guess $\xi' \in \{0, 1\}$ for the challenge bit $\xi$. Based on this, $\mathcal{B}$ outputs its decision $\mu'$ for the DBDH instance. If $\xi' = \xi$, $\mathcal{B}$ outputs $\mu' = 0$ (judging $\mathcal{Z}$ as a valid tuple); otherwise, it outputs $\mu' = 1$ (judging $\mathcal{Z}$ as a random element).

To evaluate the advantage of $\mathcal{B}$, we consider the following two cases:

When $\mu = 1$, the challenge term $\mathcal{Z}$ is a random element in the target group $\hat{\mathbb{G}}_T$. In this case, the challenge ciphertext $CT^*$ contains no effective information regarding $M_\xi$ for $\mathcal{A}$, meaning that $\xi$ is information-theoretically hidden. Thus, $\mathcal{A}$ obtains no advantage, and $\Pr[\xi' = \xi \mid \mu = 1] = 1/2$. Accordingly, the probability that $\mathcal{B}$ outputs $\mu' = \mu$ is $\Pr[\mu' = \mu \mid \mu = 1] = 1/2$.

When $\mu = 0$, the challenge term $\mathcal{Z} = \hat{e}(g_1, g_2)^{abc}$. Based on the previous analysis, $\mathcal{B}$ provides an attack environment that is statistically indistinguishable from the real scheme. If $\mathcal{A}$ breaks the scheme with a non-negligible advantage $\epsilon$, then $\Pr[\xi' = \xi \mid \mu = 0] \geq 1/2 + \epsilon$. Based on the decision logic, the probability of $\mathcal{B}$ making a correct decision is $\Pr[\mu' = \mu \mid \mu = 0] \geq 1/2 + \epsilon$.

Assuming the challenger chooses $\mu$ uniformly at random and the simulation does not abort, the overall probability that $\mathcal{B}$ makes a correct decision is as follow:

$$
\begin{aligned}
\Pr[\mu' = \mu] &= \frac{1}{2} \Pr[\mu' = \mu \mid \mu = 0] + \frac{1}{2} \Pr[\mu' = \mu \mid \mu = 1] \\
&\geq \frac{1}{2} \left( \frac{1}{2} + \varepsilon \right) + \frac{1}{2} \cdot \frac{1}{2} \\
&= \frac{1}{2} + \frac{\varepsilon}{2}
\end{aligned}
\tag{35}
$$

Thus, the advantage of $\mathcal{B}$ achieves a DBDH advantage of $\varepsilon_{\text{guess}} = \left| \Pr[\mu' = \mu] - \frac{1}{2} \right| \geq \frac{\varepsilon}{2}$.

To complete the security reduction, $\mathcal{B}$ must ensure that the simulation does not abort due to the inability to respond to the private-key queries issued by the adversary $\mathcal{A}$. Specifically, a simulation collapse occurs if the attribute combination requested by $\mathcal{A}$ allows $\mathcal{A}$ to extract secret components unknown to $\mathcal{B}$ via linear combinations. Following the combinatorial

analysis in [16], let $N_i$ denote the number of combinations that can yield valid private keys when $\mathcal{A}$ requests $i$ private keys. In the worst-case scenario, where $\mathcal{A}$ possesses $n_k - 1$ private keys controlled by authority $A_k$, the upper bound of the abort probability for a single authority $A_k$ is estimated as:

$$\Pr[Abort_k] \approx \frac{1}{r-1}\left(\frac{N_2}{C_{r-1}^2} + \cdots + \frac{N_{n_k-1}}{C_{r-1}^{n_k-1}}\right) < \frac{N_2(n_k-2)}{C_{r-1}^2} \cdot \frac{1}{r-1} < \frac{n_k-2}{(r-1)^2}$$

(36)

where $r$ is the prime order of the cyclic group, given that the number of attributes $n_k$ managed by each authority satisfies $n_k \ll r$, this abort probability is negligible in polynomial time.

Since the initialization processes of all authorities are mutually independent, the lower bound for the probability that $\mathcal{B}$ does not abort during the entire game is $\Pr[\neg Abort] = \prod_{k\in\{1,\ldots,N\}}(1 - \Pr[Abort_k])$. By combining the decision advantage derived previously, the final global advantage lower bound $\epsilon'$ for $\mathcal{B}$ to break the DBDH hardness assumption satisfies:

$$\epsilon' = \Pr[\neg Abort] \cdot \epsilon_{guess} \geq \frac{\epsilon}{2} \cdot \prod_{k\in\{1,\ldots,N\}}\left(1 - \frac{n_k-2}{(r-1)^2}\right)$$

(37)

Since the advantage $\epsilon$ of $\mathcal{A}$ is non-negligible and $n_k \ll r$, the overall success probability of the simulation remains non-negligible, establishing a valid reduction: any polynomial-time adversary with non-negligible advantage against the proposed scheme can solve the DBDH problem with non-negligible advantage, completing the proof.

**Collusion and key escrow mitigation:** The security of the BMHADS scheme is built upon a decentralized key construction logic. First, any two authorities, $A_k$ and $A_j$, collaborate to generate a PRF seed $\lambda_{kj}$, which remains strictly confidential between them. As established in our IND-CPA security proof (Theorem 1), even if $\mathcal{A}$ corrupts $N-2$ authorities, it remains unaware of the PRF seeds shared by the remaining two honest authorities, ensuring that the challenge ciphertext cannot be distinguished without solving the DBDH problem.

Furthermore, the user's secret key is constructed by incorporating the master private keys of all authorities. This cumulative structure implies that even if $N-1$ authorities exhibit malicious behavior or collude, the adversary remains unable to derive a valid, functional secret key because of the missing component provided by the single honest authority. Consequently, the scheme is designed to withstand attacks by up to $N-1$ corrupted authorities.

This architecture effectively mitigates the key escrow problem. Since complete decryption capability can be achieved only through the product of components from all $N$ independent authorities (as shown in Eq. (19)), no single authority, including the CA, possesses sufficient information to reconstruct a user's private key independently. Additionally, the anonymous key-issuing mechanism ensures that authorities do not learn the user's GID, preventing the feasible collection of user attributes through tracing. In summary, BMHADS eliminates single-point key escrow risks while safeguarding user privacy in multi-authority IoMT environments.

**Theorem 2 (Validity).** A proposed DKG protocol is valid if, under the honest majority assumption ($P \geq 3f + 1$), it effectively resists attacks from a PPT adversary $\mathcal{A}$ and satisfies the following three conditions:

(1) All honest assistant nodes obtain an identical global public key $PK_{group}$.

(2) Each honest node's local private key share $sk_u$ is valid and verifiable against public commitments.

(3) The global threshold signature $\sigma_{group}$, aggregated from valid partial signatures, is verifiable by $PK_{group}$ via bilinear pairing.

**Proof:** Consider a PPT adversary $\mathcal{A}$ that adaptively controls up to $f$ Byzantine nodes to disrupt the protocol by submitting forged shares, broadcasting inconsistent commitments, or initiating malicious complaints. Firstly, when a malicious

node $AN_u^*$ acts as a dealer, it might attempt to induce divergence in the global public key by providing inconsistent coefficient commitments $V_{u,w}^*$ to different honest nodes. However, our protocol mandates that all commitments be directly anchored on the blockchain. Leveraging the inherent consensus and immutability of the distributed ledger, honest nodes accept only the singular version of commitments ratified by the underlying BFT consensus. Consequently, for any honest node $AN_v$, the set of qualified nodes $\mathcal{H}$ and their corresponding commitments $V_{v,0}$ remain globally consistent, ensuring the uniqueness and integrity of the global public key $PK_{group} = \prod_{v \in \mathcal{H}} V_{v,0}$ (mod $r$). Secondly, $\mathcal{A}$ may attempt to disseminate forged shares $s_{u,v}^*$ that deviate from the intended polynomial distribution to undermine the algebraic integrity of subsequent signatures. Upon receiving $s_{u,v}^*$, an honest node $AN_v$ executes a Verifiable Secret Sharing (VSS) check against the commitments $V_{u,w}^*$ publicly anchored on-chain. If $\mathcal{A}$ distributes erroneous shares, the VSS verification equation $g_2 \cdot s_{u,v}^* \neq \prod_{w=0}^{t-1}(V_{u,w}^*)^{v^w}$ (mod $r$) will fail. In such an event, $AN_v$ immediately invokes a challenge mechanism on the blockchain, providing the non-compliant share and its cryptographic signature as evidence. Since on-chain commitments are immutable, the system identifies and disqualifies the malicious actor via the BFT consensus. Furthermore, the economic penalty mechanism ensures that the malicious node's deposit $Stake_u$ is forfeited, thereby significantly increasing the cost of such adversarial behavior. Finally, $\mathcal{A}$ may attempt to forge a valid threshold signature $\sigma_{group}^*$ using its $f$ compromised private key fragments. According to the Lagrange interpolation principle, reconstructing the complete threshold signature necessitates at least $t = 2f + 1$ valid partial signatures. Given that the number of nodes controlled by $\mathcal{A}$ satisfies $f < t$, and assuming the computational hardness of the DLP, $\mathcal{A}$ is incapable of deriving the missing $t - f$ secret shares from the known fragments. Therefore, as long as honest nodes generate partial signatures $\sigma_v' = sk_v \cdot h'$ in accordance with the protocol, the aggregated signature $\sigma_{group}$ will satisfy the bilinear pairing verification. $\mathcal{A}$ remains powerless to forge valid authentication credentials without reaching the requisite threshold of authorized shares.

**Theorem 3 (Robustness).** Under the honest majority assumption $P \geq 3f + 1$, the proposed DKG protocol and threshold signature scheme are robust. Specifically, in the presence of $\mathcal{A}$ controlling at most $f$ nodes, the honest nodes can still output a valid global public key $PK_{group}$ and collaboratively complete threshold-signature tasks.

**Proof:** Suppose $\mathcal{A}$ controls up to $f$ Byzantine nodes to initiate conflicts, send invalid shares during the key exchange phase, or launch silence attacks during the signature phase. To counter these threats, the protocol incorporates a BFT design requiring $P \geq 3f + 1$ assistant nodes. To ensure liveness and safety, the number of honest nodes must be at least $2f + 1$, i.e., $H \geq 2f + 1$. Even if $\mathcal{A}$ compromises $f$ nodes, it cannot achieve majority control or break the threshold $t$, as $H \geq t$ is consistently satisfied. If a node fails to respond or submits erroneous shares, an accusation protocol is triggered. Once malicious behavior is evidenced, the smart contract disqualifies the offender from the candidate set $\mathcal{H}$. Consequently, the protocol remains resilient and will not stall due to $f$ faulty nodes.

**Theorem 4 (Confidentiality).** $\mathcal{A}$ cannot obtain the system master key $SK_{group}$ during or after protocol execution.

**Proof:** The primary private key $SK_{group}$ is the sum of the polynomial constant terms $a_{u,0}$ from each assistant node. Since we use a threshold of $t = 2f + 1$ in our Byzantine-tolerant DKG protocol, an adversary $\mathcal{A}$ controlling at most $f$ malicious nodes can obtain at most $f$ partial shares $s_{u,v}$. However, reconstructing the polynomial requires at least $t = 2f + 1$ shares. Even if $\mathcal{A}$ colludes all $f$ malicious nodes, they still lack $f + 1$ additional shares from honest nodes to reach the threshold. Furthermore, the shares $s_{u,v}$ are encrypted using the recipient's public key $PK_{v,enc}$, and only authorized nodes can decrypt them. $\mathcal{A}$ cannot eavesdrop on or decrypt shares belonging to honest nodes. Finally, during the key update phase, a new polynomial $\Delta f_u(z)$ with constant term zero is generated, updating the share to $sk_v'$. Old shares become invalid, and $\mathcal{A}$ cannot infer the new share. Thus, the scheme satisfies static secrecy and forward secrecy.

**Theorem 5 (Sybil attack resistance).** $\mathcal{A}$ cannot create multiple fake identities to increase control, disrupt consensus, or compromise key generation.

**Proof:** As established in the threat model (Section 4.3), each assistant node $AN_u$ must deposit a mandatory stake $\Omega$ during the registration phase. To pose a substantive threat to the DKG process or BFT consensus, $\mathcal{A}$ must control at least $f + 1$ nodes. Consequently, the minimum financial cost $E_{total}'$ incurred by $\mathcal{A}$ is $E_{total}' = (f + 1) \cdot \Omega = E_{min}'$. Under the

assumption of a financially rational adversary, $\mathcal{A}$ will only initiate an attack if the illicit gain $E'_{\mathcal{A}}$ from compromising medical data exceeds $E'_{total}$. By setting $\Omega$ such that $E'_{min} \gg E'_{\mathcal{A}}$, the protocol renders large-scale Sybil attacks economically unfeasible. Furthermore, the blockchain ensures a unique one-to-one mapping between a node's identity and its on-chain address. If $\mathcal{A}$ utilizes its Sybil nodes to transmit inconsistent coefficient commitments or forged shares, honest nodes will generate accusation messages containing cryptographic evidence ($s_{u,v}, V_{u,w}, C_{u,v}, \sigma_{u,v}$). Upon verifying the breach via BFT consensus, the smart contract automatically executes the penalty mechanism. This results in the irreversible forfeiture of the total stake $E'_{total}$ and the immediate invalidation of all associated public keys.

## 7. Scheme analysis

In this section, we provide the theoretical and experimental analyses of the BMHADS scheme. Table 2 lists the notations employed.

### 7.1. Theoretical analysis

#### 7.1.1. Functional comparison.
To evaluate the comprehensive performance of the proposed BMHADS scheme, we compare it with several state-of-the-art schemes [18,16,15,21,33,32] in Table 3.

Regarding functional completeness, while schemes [18] and [21] support hierarchical access, they lack blockchain-based security guarantees. Schemes [15,16,21] implement multi-authority authorization, yet fall short in data source authentication and verification, which are critical in IoMT scenarios. In terms of security and traceability, only schemes [33], [32], and our proposed framework integrate blockchain technology. However, [33] and [32] fail to support complex hierarchical data sharing and multi-authority collaboration simultaneously. Notably, most existing schemes still rely on Type-1 symmetric pairings. According to modern cryptographic standards, Type-1 curves exhibit significant parameter redundancy at a 128-bit security level. In contrast, our scheme is the only one to adopt Type-3 asymmetric pairing curves while achieving all the listed security features. This choice not only enhances computational efficiency but also

**Table 2. Symbol definition.**

| Symbol | Meaning |
| --- | --- |
| $\lvert\mathbb{G}_1\rvert, \lvert\mathbb{G}_2\rvert, \lvert\hat{\mathbb{G}}_T\rvert, \lvert\mathbb{Z}_r^*\rvert$ | Size of element of in $\mathbb{G}_1$, $\mathbb{G}_2$, $\hat{\mathbb{G}}_T$, $\mathbb{Z}_r^*$ |
| $\lvert\mathbb{G}\rvert, \lvert\mathbb{G}_T\rvert$ | Size of element of in $\mathbb{G}$, $\mathbb{G}_T$ |
| $T_1, T_2, T_T$ | Exponentiation in $\lvert\mathbb{G}_1\rvert, \lvert\mathbb{G}_2\rvert, \lvert\hat{\mathbb{G}}_T\rvert$ |
| $T_G, T_{G_T}$ | Exponentiation in $\lvert\mathbb{G}\rvert, \lvert\mathbb{G}_T\rvert$ |
| $T_{typ1}$ | Time of a Type-1 bilinear pairing operation |
| $T_{typ3}$ | Time of a Type-3 bilinear pairing operation |
| $L_{ID_u}, L_{addr}$ | Lengths of node identifier and blockchain address. |
| $L_{stake}, L_{hash}$ | Lengths of node stake record and hash value. |
| $N$ | The number of authorities |
| $n_{ad}$ | Number of attributes added or removed |
| $n_{av}$ | Average number of attributes owned by users |
| $n_{re}$ | Number of attributes revoked |
| $n_T$ | Set of transport nodes |
| $n_{du}$ | The attribute set of user |
| $E_{Cl}$ | The attribute attached to the ciphertext $CT$ |
| $n_{gl}$ | Global attribute set |
| $S_l$ | The smallest internal node |

**Table 3. Comparative study of existing schemes.**

| Scheme | Hierarchical access | Multiple authorities | Collusion resistance | Blockchain | Data source authentication | Data verification | Pairing type |
|---|---|---|---|---|---|---|---|
| Scheme [18] | ✓ | × | × | × | × | × | Type-1 |
| Scheme [16] | × | ✓ | ✓ | × | × | × | Type-1 |
| Scheme [15] | × | ✓ | ✓ | × | × | × | Type-1 |
| Scheme [21] | ✓ | ✓ | ✓ | × | × | × | Type-1 |
| Scheme [33] | ✓ | × | × | ✓ | ✓ | ✓ | Type-1 |
| Scheme [32] | × | × | × | ✓ | ✓ | ✓ | Type-1 |
| Our scheme | ✓ | ✓ | ✓ | ✓ | ✓ | ✓ | Type-3 |

significantly reduces the storage overhead for resource-constrained IoMT devices. In summary, the proposed scheme outperforms existing solutions in both functional coverage and technical architecture, making it better suited to complex medical data-sharing environments.

**7.1.2. Storage overhead.** Table 4 provides a theoretical comparison of the storage overhead across various schemes, utilizing the notations defined therein to evaluate key parameters: the authority's public key ($PK_k$), the master secret key ($MSK_k$), the user's private key ($SK_U$), and the ciphertext ($CT$). A detailed analysis of the storage expressions reveals that most existing schemes [13,15,16,26,27] rely on symmetric pairings ($\hat{e} : \mathbb{G} \times \mathbb{G} \to \mathbb{G}_T$), where the storage cost is determined by the element sizes of the finite field $\mathbb{Z}_r^*$, and the groups $\mathbb{G}$ and $\mathbb{G}_T$. However, these schemes, notably [27], often employ traditional Type-1 curves that offer only 80–100 bits of security—a level insufficient for the stringent security requirements of modern IoMT environments.

In contrast, the proposed scheme utilizes asymmetric pairings ($\hat{e} : \mathbb{G}_1 \times \mathbb{G}_2 \to \hat{\mathbb{G}}_T$), delivering a standard 128-bit security level that aligns with contemporary cryptographic benchmarks. Examination of the storage expressions shows that while the overhead in [13,15,16,26,27] is primarily linearly correlated with the number of attributes ($n_{gl}$, $n_{du}$) and the size of $\mathbb{G}$ elements, the total overhead of our scheme, expressed as $(N + n_{gl})|\mathbb{G}_1| + (n_{du} + 1)|\mathbb{G}_2| + (N + n_{gl} + 1)|\mathbb{Z}_r^*|$, maintains linear growth across all terms, achieved despite the distinction between $\mathbb{G}_1$ and $\mathbb{G}_2$. In practical deployments using BLS12-381 curves, $\mathbb{G}_1$ elements are notably compact, even though $\hat{\mathbb{G}}_T$ elements are larger. Given that medical data-sharing access policies often involve a large number of attributes, the ciphertext is predominantly composed of group elements. Although the inclusion of $\hat{\mathbb{G}}_T$ introduces some overhead, it remains well within acceptable limits for most IoMT scenarios. Ultimately, the proposed scheme provides a superior security-to-overhead trade-off, making it highly suitable for resource-constrained yet security-sensitive IoMT applications.

**7.1.3. Computational overhead.** Tables 5 and 6 present a comparative analysis of the computational overhead of existing schemes and the proposed BMHADS framework, focusing on key generation, user revocation, user joining,

**Table 4. Comparison of storage overhead.**

| Scheme | $PK_k$ | $MSK_k$ | $SK_U$ | $CT$ |
|---|---|---|---|---|
| Scheme [27] | $(2n_{gl} + 1)|\mathbb{G}|$ | $(3n_{gl} + 1)|\mathbb{Z}_r^*|$ | $(2 + n_{du})|\mathbb{G}|$ | $l|\mathbb{G}_T| + 3|\mathbb{G}|(|E_{C1}| + |E_{C2}| + \cdots + |E_{Cl}|)$ |
| Scheme [26] | $N(|\mathbb{G}| + |\mathbb{G}_T|)n_{gl}$ | $4n_{gl}|\mathbb{Z}_r^*|$ | $(n_{du} + 2)|\mathbb{G}|$ | $l(|\mathbb{G}| + |\mathbb{G}_T|) + 3|\mathbb{G}|(|E_{C1}| + \cdots + |E_{Cl}|) + lT_{sym}$ |
| Scheme [16] | $N|\mathbb{G}_T| + (n_{gl} + N)|\mathbb{G}|$ | $(n_{gl} + N^2 + N)|\mathbb{Z}_r^*|$ | $(n_{du} + 1)|\mathbb{G}|$ | $l(|\mathbb{G}| + |\mathbb{G}_T|) + (|E_{C1}| + |E_{C2}| + \cdots + |E_{Cl}|)|\mathbb{G}|$ |
| Scheme [15] | $(2|\mathbb{G}| + |\mathbb{G}_T|)n_{gl} + |\mathbb{G}|$ | $(3n_{gl} + 1)|\mathbb{Z}_r^*|$ | $3n_{du}|\mathbb{G}|$ | $l|\mathbb{G}_T| + (3|\mathbb{G}| + |\mathbb{G}_T|)(|E_{C1}| + \cdots + |E_{Cl}|)$ |
| Scheme [13] | $(2 + n_{gl})|\mathbb{G}| + 2|\mathbb{G}_T|$ | $4|\mathbb{Z}_r^*|$ | $(2 + 2n_{du})|\mathbb{G}|$ | $lT_{sym} + l(2|\mathbb{G}| + |\mathbb{G}_T|) + |\mathbb{G}|(|E_{C1}| + \cdots + |E_{Cl}|)$ |
| Our scheme | $N|\hat{\mathbb{G}}_T| + (N + n_{gl})|\mathbb{G}_1|$ | $(N + n_{gl} + 1)|\mathbb{Z}_r^*|$ | $(n_{du} + 1)|\mathbb{G}_2|$ | $l(|\hat{\mathbb{G}}_T| + |\mathbb{G}_1|) + 2|E_{C1}|\mathbb{G}_1| + j'n_T|\hat{\mathbb{G}}_T|$ |

**Table 5. Execution time of cryptographic operations.**

| Scheme | KeyGen | User revocation | User joining | Policy update |
|---|---|---|---|---|
| Scheme [11] | $5n_{du}T_G + 4T_G$ | — | — | — |
| Scheme [12] | $5n_{du}T_G + 2T_G$ | 0 | $2T_G + 5n_{av}T_G$ | — |
| Scheme [10] | $4n_{du}T_G$ | — | — | — |
| Our scheme | $Nn_{du}T_2 + 3N(N-1)T_2$ | $2n_{re}T_1 + n_{re}T_2$ | $n_{av}T_2 + 6N(N-1)T_2$ | $T_1 n_{ad}$ |

**Table 6. Comparision of computation overhead.**

| Scheme | Data encryption | Data decryption |
|---|---|---|
| Scheme [27] | $lT_{G_T} + lT_{sym} + 5(|E_{C1}| + |E_{C2}| + \cdots + |E_{Cl}|)T_G$ | $ln_{du}(3T_{typ1} + T_{G_T}) + (|S_1| + |S_2| + \cdots + |S_l|)T_{G_T} + lT_{sym}$ |
| Scheme [26] | $lT_{sym} + l(T_G + T_{G_T}) + 5(|E_{C1}| + \cdots + |E_{Cl}|)T_G$ | $ln_{du}T_{typ1} + T_{G_T}(|S_1| + |S_2| + \cdots + |S_l|) + lT_{sym}$ |
| Scheme [16] | $l(T_G + T_{G_T}) + (|E_{C1}| + |E_{C2}| + \cdots + |E_{Cl}|)T_G$ | $(|S_1| + \cdots + |S_l| + l)T_{G_T} + Nn_{du}T_{typ1} + lT_{typ1}$ |
| Scheme [15] | $lT_{G_T} + (4T_G + 2T_{G_T})(|E_{C1}| + \cdots + |E_{Cl}|)$ | $(|S_1| + \cdots + |S_l|)T_{G_T} + 4ln_{du}T_{typ1}$ |
| Scheme [13] | $lT_{sym} + l(2T_G + T_{G_T}) + T_G(|E_{C1}| + \cdots + |E_{Cl}|)$ | $2l(n_{du} + 1)T_{typ1} + (|S_1| + \cdots + |S_l|)T_{G_T} + lT_{sym}$ |
| Our scheme | $lT_{sym} + l(T_1 + T_T) + 2|E_{C1}|T_1 + j'n_T T_T$ | $(Nn_{du} + l)T_{typ3} + (|S_1| + 2l + Nj''n_T)T_T + lT_{sym}$ |

policy updates, and encryption/decryption. The evaluation focuses exclusively on the execution time of exponentiation operations across groups $\mathbb{G}$, $\mathbb{G}_1$, $\mathbb{G}_2$, $\mathbb{G}_T$, and $\hat{\mathbb{G}}_T$, as well as the bilinear pairing operations $T_{typ1}$ and $T_{typ3}$.

As illustrated in Table 5, the key generation cost in schemes [10,11,12] correlates primarily with the number of user attributes. In contrast, our proposed scheme exhibits a linear relationship with both the number of authorities and the attributes possessed by a user. Regarding user revocation in scheme [12], the algorithm involves the cloud server removing corresponding user entries from a key list; since this primarily entails storage deletion, its computational overhead for user departure is negligible (zero). However, the proposed scheme incurs a computational cost of $2n_{re}T_1 + n_{re}T_2$ for user revocation, where the complexity scales linearly with the number of revoked users ($n_{re}$) to maintain decentralized security. For user joining, the computational cost of scheme [12] is $2T_G + 5n_{av}T_G$, whereas our scheme requires $n_{av}T_2 + 6N(N-1)T_2$. This overhead arises because users must interact with multiple authorities via anonymous protocols to ensure decentralized trust. Furthermore, while the compared schemes do not address policy updates, our framework minimizes this cost by only updating modified attributes, resulting in an efficient overhead of $T_1 n_{ad}$.

As shown in Table 6, in schemes such as [13,15,16,26,27], data owners must generate independent access policies and perform separate encryption operations for each of the $l$ files. Consequently, the encryption complexity is determined by the cumulative set of attributes across all files, denoted as $\{E_{c1}, \ldots, E_{cl}\}$. Similarly, during decryption, users must simultaneously satisfy $l$ distinct access policies, making the complexity proportional to the set of smallest internal nodes $\{|S_1|, \ldots, |S_l|\}$ across all structures. This results in a computational burden that scales linearly with the file count $l$. In contrast, the proposed approach introduces an integrated hierarchical access structure. By unifying access control for multiple files into a single policy, our design significantly reduces redundant computations. This one-time encryption for multi-level files eliminates the need for repetitive policy processing, thereby substantially lowering overall computational overhead and enhancing system efficiency.

**7.1.4. Complexity of the DKG Protocol.** The formal communication complexity of the DKG protocol is modeled by analyzing the efficiency of its interactions across its entire lifecycle. In the registration phase, each assistant node broadcasts its identity $ID_u$ and dual public key pairs ($PK_{u,sign} \in \mathbb{G}_1$, $PK_{u,enc} \in \mathbb{G}_2$), incurring a linear communication overhead of $O(P)$. During the commitment phase, the blockchain BFT consensus acts as a reliable broadcast channel for nodes to upload commitment vectors $\{V_{u,w}\}_{w=0}^{t-1} \in \mathbb{G}_2$, resulting in a complexity of $O(P^2 + Pt)$ derived from consensus

interactions and payload weight. The share exchange phase involves $P(P-1)$ point-to-point transmissions of encrypted fragments $C_{u,v}$, yielding $O(P^2)$ complexity, while the complaint phase (if triggered) similarly requires $O(P^2)$ for evidence synchronization via BFT consensus. Furthermore, the collaborative authentication phase entails collecting $P$ partial signatures ($O(P)$) and broadcasting a constant-size aggregate signature ($O(1)$). Finally, the key update phase generates $O(P^2)$ traffic by distributing encrypted share increments $\Delta s_{u,v}$. Consequently, the overall communication complexity of the protocol is formally defined as $O(P^2)$.

**On-chain storage overhead analysis.** Let $P$ denote the number of assistant nodes, $t$ the secret sharing threshold, and $f$ the maximum number of tolerable malicious nodes satisfying $P \geq 3f + 1$. The on-chain storage overhead of the proposed BMHADS scheme consists of four primary components.

First is the node registration phase, where each assistant node persists its unique identifier of $L_{ID_u}$ bytes, blockchain address of $L_{addr}$ bytes, encryption public key of $|PK_{enc,u}|$ bytes, signing public key of $|PK_{sign,u}|$ bytes, and security deposit of $L_{stake}$ bytes on the ledger. This results in a constant per-node registration storage cost $\mathcal{O}_{reg} = L_{ID_u} + L_{addr} + |PK_{enc,u}| + |PK_{sign,u}| + L_{stake}$ and a cumulative network load of $P \cdot \mathcal{O}_{reg}$.

Subsequently, during the commitment phase, each node broadcasts a commitment vector of length $t$ in group $\mathbb{G}_2$ to facilitate VSS, generating a core storage load of $P \cdot t \cdot |\mathbb{G}_2|$ bytes.

Additionally, the complaint and arbitration phase generates on-demand overhead, as a single evidence package comprises the accused node's address, invalid share ciphertexts, local signatures, accuser signatures, and corresponding commitments. The storage cost for a single complaint is defined as $\mathcal{O}_{complaint} = L_{addr} + |C_{u,v}| + |\sigma_{u,v}| + t \cdot |\mathbb{G}_2| + |\sigma_{auth}|$, leading to a total worst-case overhead of $f \cdot \mathcal{O}_{complaint}$, though the impact of such rare events on long-term ledger expansion remains limited.

Finally, in the key update and authentication phase, the system persists the hash evidence of new shares of $P \cdot L_{hash}$ bytes along with a constant-size system-level aggregate signature of $|\sigma_{group}|$ bytes.

Consequently, the expression for the total on-chain storage overhead is $\mathcal{O}_{total} = P \cdot \mathcal{O}_{reg} + P \cdot t \cdot |\mathbb{G}_2| + f \cdot \mathcal{O}_{complaint} + P \cdot L_{hash} + |\sigma_{group}| + \mathcal{O}_{meta}$, where $\mathcal{O}_{meta}$ represents the auxiliary system metadata overhead required by the blockchain infrastructure, including transaction headers, timestamps, and block index information.

## 7.2. Experimental analysis

The proposed BMHADS scheme is implemented in C/C++ using the MIRACL cryptographic library within the Visual Studio Code environment. To provide a comprehensive evaluation, we implement both Type-1 symmetric pairings (based on a supersingular curve) and Type-3 asymmetric pairings (based on the BLS12–381 curve) to achieve 128-bit security. Additionally, the blockchain infrastructure is deployed on the Hyperledger Fabric 2.4 consortium platform with a BFT consensus mechanism, ensuring consistent synchronization of DKG commitments and decentralized evidence storage for collaborative authentication. The specific hardware configurations, blockchain parameters, and detailed cryptographic parameter settings used in our experiments are summarized in Table 7.

In the experimental configuration, the number of AAs is fixed at $N = 10$ to simulate a large-scale decentralized healthcare network. For the access structure, hybrid threshold gates are employed to evaluate system performance under complex policies, while the transmission node scales $n_T$ and their corresponding subnodes $j'$ are kept within a small range. Two main testing scenarios are implemented. First, the file hierarchy depth is fixed at $l = 5$, and the number of attributes in the set $Att(x,y)$ is varied across {10, 20, 30, 40, 50}. Second, the attribute count is fixed at 20, and the number of files $l$ is varied across {4, 6, 8, 10, 12} to assess the efficiency of hierarchical processing.

Fig 6(a) compares the ciphertext storage overhead of different CP-ABE schemes under varying numbers of attributes. The storage cost of all schemes increases with the number of attributes. Among them, the scheme in [15] shows the fastest growth, followed by the schemes in [26,27], while the schemes in [13,16] exhibit lower growth. For attribute counts ranging from 10 to 50, the ciphertext storage overhead of the proposed scheme remains lower than that of all comparison

**Table 7. Detailed experimental settings and parameters (Including Blockchain).**

| Category | Type-1 Symmetric Pairing | Type-3 Asymmetric Pairing |
|---|---|---|
| **Hardware** | Intel Core i7-9750H CPU @ 2.60GHz, 8GB RAM | |
| **Software** | VS Code, C/C++, MIRACL, Hyperledger Fabric 2.4 | |
| **Blockchain Parameters** | | |
| Consensus Mechanism | Byzantine Fault Tolerant (BFT) | |
| Block Size | 2 MB | |
| Block Interval | 2 s | |
| Total Node Count ($P$) | 100 | |
| Max Malicious Nodes ($f$) | 33 ($P \geq 3f + 1$) | |
| **Cryptographic Settings** | | |
| Curve Type | Super-singular Curve | BLS12–381 Curve |
| Curve Equation | $y^2 = x^3 - 3x$ | $y^2 = x^3 + 4$ |
| Embedding Degree ($k$) | 2 | 12 |
| Base Field Modulus ($q$) | 1536 bits | 381 bits |
| Group Order ($r$) | 256 bits | 255 bits |
| Security Level | 128-bit | 128-bit |
| **Element Sizes** | | |
| Size of Base Group 1 ($\mathbb{G}$ or $\mathbb{G}_1$) | 3072 bits | 768 bits |
| Size of Base Group 2 ($\mathbb{G}$ or $\mathbb{G}_2$) | 3072 bits | 1536 bits |
| Size of Target Group ($\mathbb{G}_T$ or $\hat{\mathbb{G}}_T$) | 3072 bits | 4608 bits |
| Scalar Space $\mathbb{Z}_r^*$ | 256 bits | 255 bits |

schemes. Existing schemes require embedding the entire access policy into the ciphertext, resulting in significant redundancy, whereas the proposed scheme effectively reduces ciphertext storage overhead by leveraging a hierarchical structure.

Fig 6(b) compares the key generation time of different CP-ABE schemes under varying numbers of attributes. As shown, the computational cost of key generation increases linearly with the number of attributes. In the proposed BMHADS scheme, the number of attribute authorities is fixed at $N = 10$. When the number of attributes is less than 20, the key generation time of our scheme is slightly higher than that of scheme [10]. When the attribute count ranges from 10 to 12, our scheme also exhibits slightly higher key generation time than schemes [11,12]. However, when the attribute count exceeds 20, the key generation time of our scheme is lower than that of schemes [10,11,12], demonstrating superior performance.

Fig 6(c) compares the computational overhead of different CP-ABE schemes in terms of encryption time as the number of files increases. The encryption overhead increases linearly with the number of attributes. The encryption overhead of scheme [15] is significantly higher than that of the other schemes. The three schemes [13,26,27] are comparable. Scheme [16] has lower overhead than these, but higher than that of the proposed scheme. The proposed scheme has the lowest overhead and the slowest growth rate due to the reduction in redundant attribute encryption achieved by its hierarchical structure.

Fig 6(d) compares the computational cost of decryption as the number of files increases. The decryption overhead for each scheme increases linearly with the number of attributes. Scheme [15] has the highest cost, while [27] and [13] are similar and rank second, and [26] and [16] are similar and lower. The proposed scheme consistently has the lowest cost when decrypting 4–12 nested files.

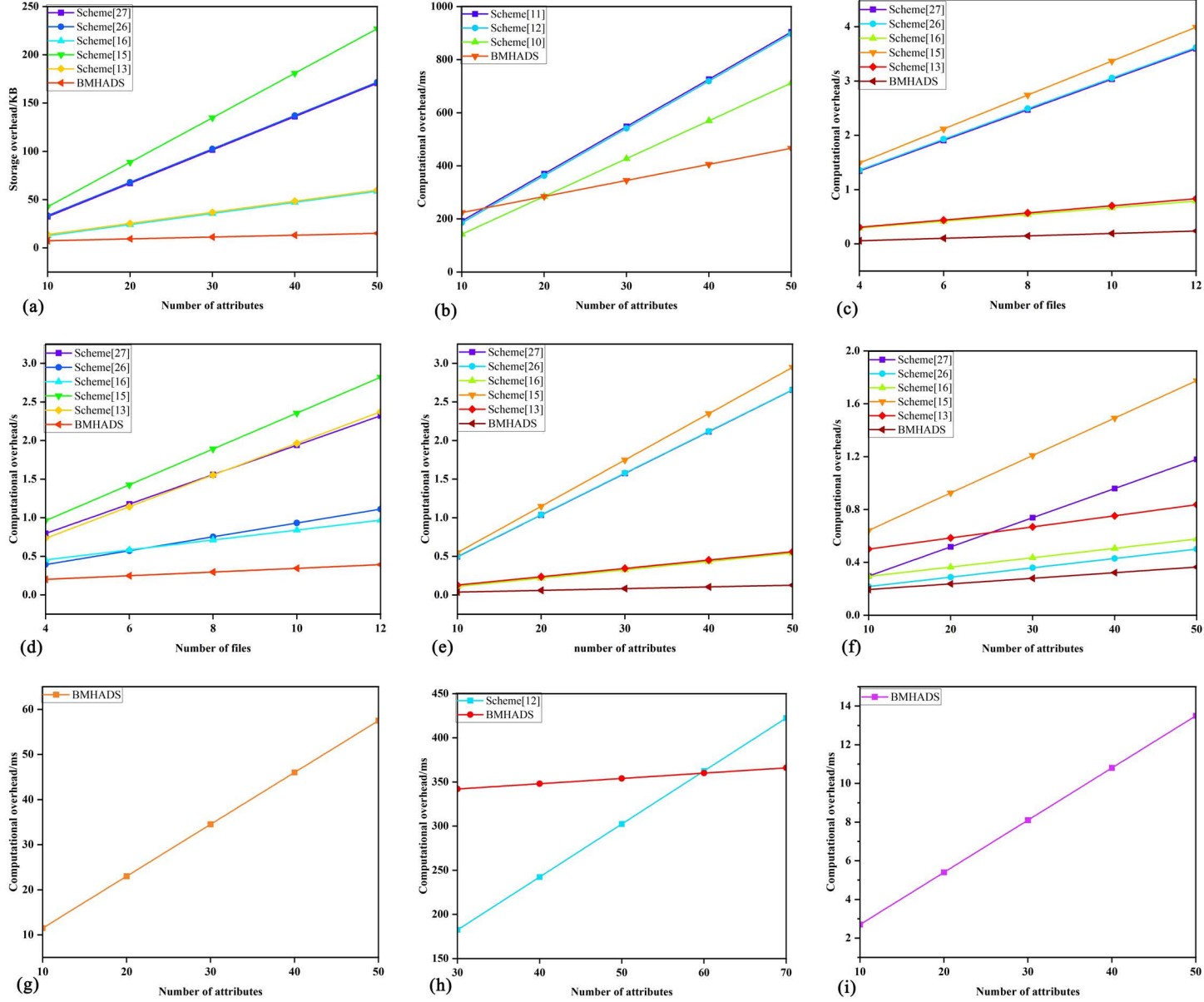

**Fig 6. Performance evaluation of the proposed BMHADS scheme and existing schemes.** (a) Ciphertext size, (b) Key generation, (c) Encryption files, (d) Decryption files, (e) Encryption attributes, (f) Decryption attributes, (g) User revocation, (h) User joining, (i) Policy update.

Fig 6(e) compares the computational overhead of encryption time as the number of attributes increases. The overhead of all schemes increases linearly. The schemes in [15,26,27] increase more rapidly, while those in [13,16] increase more slowly. The computational cost of the proposed scheme is lower than that of [13,16]. Therefore, for a fixed number of files, the proposed scheme always has the lowest encryption cost.

As shown in Fig 6(f), the decryption time for all schemes increases with the number of attributes. Scheme [15] has the highest overhead. When the number of attributes is less than 20, scheme [13] has a higher overhead than scheme [27]. When the number of attributes is approximately 25, scheme [13] has lower overhead than scheme [27]. Schemes [26] and

[16] have lower decryption times, with scheme [26] performing slightly better than scheme [16]. The proposed scheme, BMHADS, maintains the lowest decryption cost across all attribute counts.

Fig 6(g) shows that the computational overhead of user revocation in the proposed scheme increases linearly with the number of attributes, due to the proposed scheme's use of a proxy-based re-encryption key-update mechanism, which requires re-encrypting only the affected ciphertext components.

Fig 6(h) compares the cost of user enrollment when the number of authorities is fixed at 10. When the number of attributes is less than 60, the cost of scheme [12] is lower than that of BMHADS, because BMHADS incurs a fixed overhead due to the execution of the anonymous exchange protocol. When the number of attributes exceeds 60, the cost of BMHADS is lower than that of scheme [12].

Fig 6(i) shows that the policy update overhead increases linearly with the number of attributes. The proposed BMHADS scheme updates only the portions of the attribute ciphertext that have changed, leaving the rest unchanged, thereby reducing policy update costs.

**Blockchain Experiment**: To simulate realistic IoMT scenarios and evaluate the system's performance at scale, the number of assistant nodes is fixed at $P = 100$, and the corresponding threshold is set to $t = [\frac{2}{3}P]$. Table 8 provides a comprehensive decomposition of the computational time, communication overhead, and blockchain interaction costs under this configuration.

As illustrated in Fig 7(a), we evaluated the evolution of storage overhead as the number of nodes ($P$) scaled from 10 to 100. Experimental results demonstrate that the storage load increases from 16.97 KB at $P = 10$ to 1293.8 KB (1.27 MB) at $P = 100$. Although the growth follows a quadratic trend, the overhead remains well below the single-block capacity limit, verifying the system's scalability in large-scale healthcare collaborative environments.

Fig 7(b) illustrates that the aggregate threshold signature maintains a fixed length of 96 bytes and constant space complexity. Traditional ECDSA-based schemes, by contrast, exhibit linear overhead growth relative to the number of participating nodes. Substantial storage optimization curtails the on-chain footprint and eases the communication load during blockchain consensus. Empirical measurements confirm that 100 assistant nodes complete the end-to-end key update activation within 1.52 s. Rapid state transitions ensure scalable deployment across large-scale IoMT networks involving resource-limited devices.

Based on the consensus latency trends illustrated in Fig 7(c), the sub-second delay exhibited by the system has a negligible impact on the real-time availability of medical data. The BMHADS framework achieves deep decoupling between the control plane and the data plane, with the encryption and transmission of real-time physiological indicators treated

**Table 8.** Comprehensive performance decomposition and system costs ($P = 100$, $t = 67$).

| Metric and Protocol Operation | Entity | Complexity | Measured Value |
| --- | --- | --- | --- |
| **Computation Time (ms)** | | | |
| DKG Share Refreshment (Parallel) | ANs | $O(t \cdot T_2)$ | 81.08 ms |
| Threshold Signature Aggregation | ANs | $O(P \cdot T_1)$ | 18.51 ms |
| **Communication Cost (MB)** | | | |
| DKG Commitment Broadcast | Network | $O(P \cdot t)$ | 1.23 MB |
| Encrypted Share Distribution | Network | $O(P^2)$ | 2.14 MB |
| **Blockchain Interaction** | | | |
| On-chain Storage Increment | Ledger | $O(P \cdot t)$ | 1.27 MB |
| Consensus Latency (at 300 TPS) | ANs | $O(P^2)$ | 280.00 ms |
| Global state synchronization delay | ANs | N/A | 1.52 s |

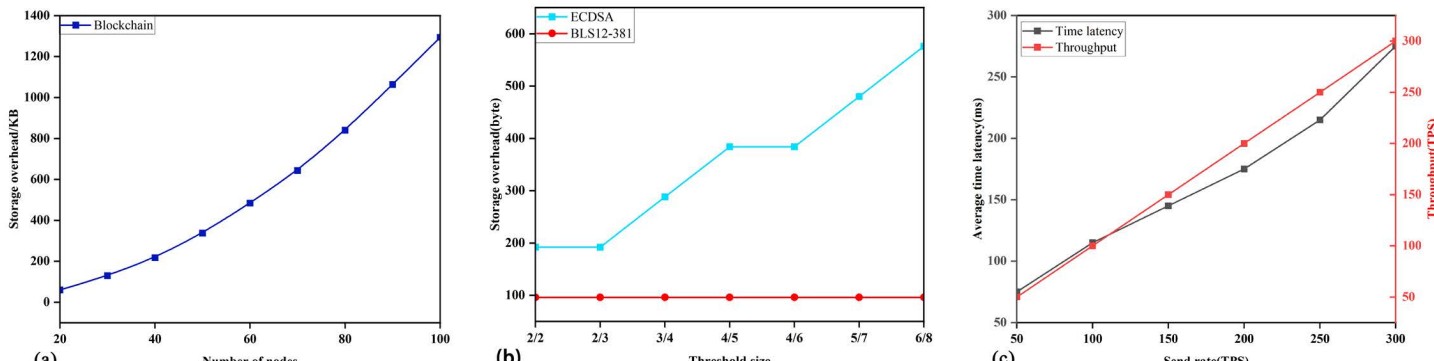

**Fig 7. Experimental evaluation of blockchain storage: overhead, signature, and transaction performance.** (a) Blockchain overhead, (b) Signature overhead, (c) Latency & throughput.

as data-plane operations that bypass the blockchain consensus process. The availability of these high-frequency data streams is primarily determined by local computational efficiency and network bandwidth, with the measured encryption time increasing from 36.5 ms (for 10 attributes) to 126.1 ms (for 50 attributes), demonstrating linear scalability with respect to the attribute set size. In contrast, the blockchain serves solely as the control plane for low-frequency security tasks, such as DKG commitment synchronization and key updates. This architectural paradigm ensures that the 280 ms consensus latency introduces minimal authorization delay and does not obstruct the continuous ingestion or authorized retrieval of critical clinical monitoring flows. Consequently, the results confirm that the proposed scheme fulfills the stringent responsiveness requirements of emergency medical scenarios while maintaining strong consistency in security governance.

**BFT Experiment**: As illustrated in Fig 8(a), we analyzed the impact of varying malicious node ratios *f* on the source authentication success rate. The results demonstrate that the system maintains a 100% authentication success rate

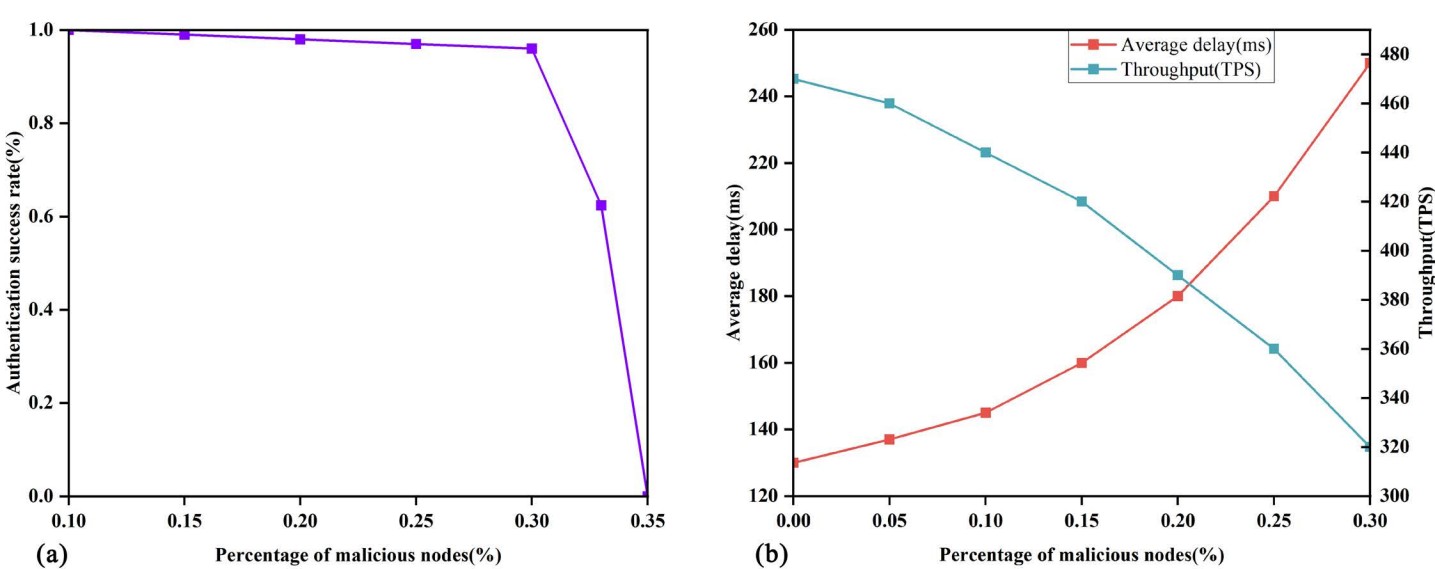

**Fig 8. Blockchain authentication rate and performance.** (a) Authentication success rate, (b) Performance comparison.

provided that *f* remains below the theoretical BFT threshold $f < 33\%$, whereas the success rate drops abruptly to zero once *f* exceeds this limit. These findings underscore the critical role of BFT in safeguarding the authenticity of medical data. In multi-authority collaborative healthcare environments, BFT consensus ensures that honest nodes reach a consistent, correct agreement even if a fraction of ANs are compromised by attackers attempting to inject fraudulent physiological data. Such resilience effectively mitigates the risk of clinical misdiagnosis arising from data tampering, providing a trustworthy and high-fidelity foundation for patient diagnostics.

Fig 8(b) further quantifies the additional performance overhead caused by Byzantine attacks: experimental results show that for every 5% increase in the proportion of malicious nodes, consensus latency increases by approximately 15–25 ms. In an extreme attack scenario where malicious nodes account for 30% of the network, system consensus latency rises to 250 ms due to message retransmissions and multi-round broadcast verification. Analysis indicates that, even under such hostile network conditions, a 250 ms delay remains well below the widely accepted 1-second real-time response threshold required for clinical medical monitoring. Furthermore, thanks to the architecture's decoupling of the control plane from the data plane, high-frequency, real-time physiological data is transmitted via an ultra-fast off-chain algorithm that performs data anonymization in only 36.5 ms-126.1 ms, with availability completely independent of the on-chain consensus process. This design ensures that performance fluctuations in the consensus layer affect only low-frequency management operations, such as permission synchronization, and never block the core real-time monitoring stream. It strongly demonstrates the feasibility of deploying this solution and its ability to ensure business continuity in real-world IoMT environments where resource-constrained endpoints coexist with complex network threats.

The computational offloading model validates the practical deployment feasibility of BMHADS on resource-constrained IoMT hardware. By delegating intensive tasks, including BFT consensus interactions and DKG maintenance, to capable assistant nodes, terminal endpoints, such as wearable sensors, can execute millisecond-level local operations, such as BLS signing, in approximately 0.27 ms while maintaining a minimal local storage footprint. This architecture prioritizes battery longevity and storage efficiency while ensuring that resource-limited devices maintain seamless business continuity even under complex network threats.

## 8. Conclusion

This study proposes a BMHADS framework for the IoMT. The framework implements fine-grained access control through a hierarchical CP-ABE and utilizes blockchain-based distributed threshold signatures to enhance system integrity and traceability. The proposed DKG protocol satisfies validity, robustness, confidentiality, and Sybil attack resistance. A key update mechanism based on PRE supports efficient user revocation. Security analysis demonstrates that the system can resist collusion among multiple authorizing entities and chosen-plaintext attacks; experimental results validate its superior computational and storage efficiency compared to existing solutions. Future work will incorporate an edge computing architecture to offload computationally intensive tasks, such as the hierarchical CP-ABE algorithm, to edge nodes. It will investigate lightweight consensus mechanisms and distributed caching strategies further to enhance the system's real-time performance and applicability.

## Supporting information

**S1 Appendix. Source code.** All code used in this study is available in the following public GitHub repositories. • MIRACL Main Repository: https://github.com/mirac1/MIRACL. • MIRACL Core Library: https://github.com/mirac1/core. (PDF)

**S1 File. Minimal dataset.** This file contains the minimal anonymized dataset underlying the findings presented in this study. (XLSX)

## Author contributions

**Conceptualization:** Hao Yuan.

**Formal analysis:** Hao Yuan.

**Methodology:** Hao Yuan.

**Supervision:** Guofang Dong.

**Validation:** Leilei Zhao.

**Writing – original draft:** Hao Yuan.

**Writing – review & editing:** Guofang Dong.

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
