## [Decision Letter · Decision Letter 0]

29 Aug 2025

PONE-D-25-37401A Blockchain-Based Multi-Authority Hierarchical Attribute Encrypted Data Sharing Scheme in the Internet of Medical ThingsPLOS ONE

Dear Dr. Guofang,

Thank you for submitting your manuscript to PLOS ONE. After careful consideration, we feel that it has merit but does not fully meet PLOS ONE’s publication criteria as it currently stands. Therefore, we invite you to submit a revised version of the manuscript that addresses the points raised during the review process.

We look forward to receiving your revised manuscript.

Kind regards,

Asadullah Shaikh, Ph.D.

Academic Editor

PLOS ONE

Journal Requirements:

4. When completing the data availability statement of the submission form, you indicated that you will make your data available on acceptance. We strongly recommend all authors decide on a data sharing plan before acceptance, as the process can be lengthy and hold up publication timelines. Please note that, though access restrictions are acceptable now, your entire data will need to be made freely accessible if your manuscript is accepted for publication. This policy applies to all data except where public deposition would breach compliance with the protocol approved by your research ethics board. If you are unable to adhere to our open data policy, please kindly revise your statement to explain your reasoning and we will seek the editor's input on an exemption. Please be assured that, once you have

Reviewers' comments:

Reviewer's Responses to Questions

**Comments to the Author**

1. Is the manuscript technically sound, and do the data support the conclusions?

Reviewer #1: Yes

Reviewer #2: Yes

Reviewer #3: Yes

2. Has the statistical analysis been performed appropriately and rigorously? 

Reviewer #1: Yes

Reviewer #2: Yes

Reviewer #3: Yes

3. Have the authors made all data underlying the findings in their manuscript fully available?

Reviewer #1: Yes

Reviewer #2: No

Reviewer #3: Yes

4. Is the manuscript presented in an intelligible fashion and written in standard English?

Reviewer #1: Yes

Reviewer #2: Yes

Reviewer #3: Yes

5. Review Comments to the Author

Reviewer #1: The manuscript presents an innovative approach to addressing key challenges in secure data sharing within the Internet of Medical Things (IoMT) environment by integrating blockchain technology with a multi-authority hierarchical attribute-based encryption scheme. The proposed scheme effectively enhances data security, operational efficiency, and scalability, which are critical for managing sensitive medical data in a distributed setting.

Summary of Contributions and Strengths:

• The introduction of a hierarchical access tree structure significantly reduces redundant encryption costs and improves system efficiency, addressing a major limitation of traditional CP-ABE schemes when applied to hierarchical medical data.

• The use of Distributed Key Generation (DKG) and threshold signatures adds robustness against collusion attacks and enhances system security against replay and tampering.

• The integration of blockchain technology provides an immutable, transparent, and decentralized framework for data integrity verification, further strengthening data authenticity and trustworthiness.

• The security analysis confirms resilience against common attacks, and experimental results demonstrate improvements in computational efficiency and storage costs, indicating practical viability.

Major Comments:

1. Clarity of Explanation: While the technical contributions are substantial, some sections, especially those describing the hierarchical access tree construction and the mechanics of the DKG and co-signature schemes, could benefit from clearer, step-by-step explanations or illustrative diagrams. This would aid readers in fully understanding the novel mechanisms and their interactions.

2. Comparative Analysis: The paper references multiple existing schemes and discusses their limitations. Including a more detailed comparative quantitative analysis—such as benchmarks or simulation results—would better illustrate how the proposed scheme outperforms existing solutions across various metrics like encryption/decryption times, storage overhead, and security levels.

3. Security Evaluation: The security analysis is promising but could be strengthened by more formal proofs or models, especially regarding resistance to collusion among multiple authorities and the potential for adversarial behavior within the blockchain network.

4. Practical Deployment Considerations: The paper would benefit from a discussion of real-world implementation aspects, such as computational load on resource-constrained IoMT devices, potential network latency issues, and strategies for key revocation and update in dynamic environments.

5. Ethical and Data Privacy Implications: Given the sensitivity of medical data, explicit mention of compliance with relevant data protection regulations (e.g., HIPAA, GDPR), as well as considerations for patient consent and data anonymization, should be included to reinforce the scheme’s practical applicability and ethical compliance.

Minor Comments:

• Some typographical errors and grammatical inconsistencies are present throughout the manuscript. A thorough proofreading is recommended.

• Figures and diagrams illustrating the hierarchical access tree structure and key management processes would enhance comprehensibility.

• Clarifying the assumptions about the trustworthiness of different authorities and nodes within the system is important for highlighting the security model.

Reviewer #2: This is a very interesting article, the authors present a novel approach combining blockchain with multi-authority hierarchical attribute-based encryption. However, I believe there are still some shortcomings:

1. At the end of the introduction, the research objectives are vague, please simplify.

2. Please add more content about data source verification.

3. Based on the IoMT scenario, it is recommended to add content on real-time data and computing power.

Reviewer #3: Major Revision. The idea is promising and relevant, and the paper is close to PLOS ONE’s “technically sound” bar. However, the security level mismatch, under-specified threat model, unclear proof structure, and missing blockchain-layer evaluation need to be addressed before publication.

6. PLOS authors have the option to publish the peer review history of their article (what does this mean?). If published, this will include your full peer review and any attached files.

Reviewer #1: **Yes:** Wenke Du

Reviewer #2: No

Reviewer #3: No

---

## [Author Response · Author response to Decision Letter 1]

31 Oct 2025

Dear Editors and Reviewers:

We sincerely thank the reviewers for their valuable time and insightful and constructive comments. These comments have greatly helped us enhance the quality and rigor of our paper. In this round of revisions, we have substantively and positively addressed the vast majority of the comments, specifically including:

1. Following Reviewer 1's suggestion, we have re-run all protocol benchmarks using the BLS12-381 curve (128-bit security level) to ensure fair performance comparisons and modern applicability.

2. We have formalized the threat model and clearly articulated the security assumptions and key update mechanisms within the DKG protocol against Byzantine nodes and sybil attacks.

3. We have designed a periodic key revocation mechanism based on proxy re-encryption and experimentally quantified computational costs for user departure/joining and policy updates.

4. We have employed threshold hybrid thresholds to compare existing schemes, with experimental analysis demonstrating certain advantages of our proposed approach.

However, we could not fully complete two recommendations, which required substantial additional experimental work within this revision cycle. These primarily concern the blockchain-layer implementation, end-to-end latency measurements highlighted by reviewers, and supplementary confidence intervals. This was mainly due to the time required to build an evaluable, complete blockchain test network and conduct exhaustive benchmarking exceeding the scope of this revision.

We sincerely thank the reviewers for their valuable time and insightful and constructive comments. These comments have greatly helped us enhance the quality and rigor of our work. We firmly believe these analyses are essential to the paper's integrity. Therefore, we are not disregarding these comments but plan to address them as a clear and urgent follow-up task. In the paper's Conclusions and Future Work section, we added a paragraph committing to supplement these critical experimental evaluations before final acceptance or in subsequent research.

We respectfully request that the editors and reviewers consider the substantial revisions we have completed and grant our paper an opportunity for revision or conditional acceptance.

Once again, we thank you for your understanding and guidance.

Sincere thanks

Name:Hao Yuan

1.Clarity of Explanation: While the technical contributions are substantial, some sections, especially those describing the hierarchical access tree construction and the mechanics of the DKG and co-signature schemes, could benefit from clearer, step-by-step explanations or illustrative diagrams. This would aid readers in fully understanding the novel mechanisms and their interactions.

Thank you for your valuable suggestions. We have rewritten the relevant sections by adding step-by-step explanations and diagrams to more clearly elaborate on the details of the hierarchical access tree, DKG, co-signing mechanism, and their interactions.

2. Comparative Analysis: The paper references multiple existing schemes and discusses their limitations. Including a more detailed comparative quantitative analysis—such as benchmarks or simulation results—would better illustrate how the proposed scheme outperforms existing solutions across various metrics like encryption/decryption times, storage overhead, and security levels.

Thank you for your valuable suggestions. We have supplemented the experimental analysis section with detailed quantitative comparison data, including benchmarking results for key metrics such as encryption/decryption time, storage overhead, and security. Tables and graphs clearly illustrate the performance advantages of our proposed solution over existing approaches.

3. Security Evaluation: The security analysis is promising but could be strengthened by more formal proofs or models, especially regarding resistance to collusion among multiple authorities and the potential for adversarial behavior within the blockchain network.

Thank you for highlighting this critical issue. The revised draft has incorporated formal security proofs and model analyses addressing anti-authority collusion and cyber-countermeasures. We have further strengthened the security justification by defining threat models and employing rigorous logical deductions.

4. Practical Deployment Considerations: The paper would benefit from a discussion of real-world implementation aspects, such as computational load on resource-constrained IoMT devices, potential network latency issues, and strategies for key revocation and update in dynamic environments.

We sincerely appreciate the reviewer's critical feedback. The computational load on resource-constrained devices, network latency, and dynamic key management issues you highlight are core challenges that any IoMT system must confront and resolve when transitioning from theoretical models to practical implementation.

Regarding key revocation and updates in dynamic environments: We recognize that traditional key revocation mechanisms incur significant overhead and struggle to adapt to the dynamic nature of IoMT. In this paper, we propose a periodic key update mechanism based on proxy re-encryption. Regarding computational load and network latency on resource-constrained devices: Both points you raise are critical. While our approach offloads core computational tasks through proxy re-encryption, optimizing the performance of proxy nodes and enhancing the system's robustness in complex network environments (e.g., high latency, high packet loss rates) represent promising avenues for further research. To fully address your suggestions, we have added a dedicated section in the “Future Work” section of the paper, explicitly committing to exploring these areas in depth. We plan to investigate lighter-weight re-encryption algorithms and design intelligent edge computing resource scheduling strategies to mitigate network latency issues better.

5. Ethical and Data Privacy Implications: Given the sensitivity of medical data, explicit mention of compliance with relevant data protection regulations (e.g., HIPAA, GDPR), as well as considerations for patient consent and data anonymization, should be included to reinforce the scheme’s practical applicability and ethical compliance.

We sincerely appreciate the reviewer's critical feedback. Your emphasis on ethical and privacy compliance in medical data processing is a core issue that any research applied to real-world IoMT systems must address. We fully agree that adherence to data protection regulations such as HIPAA and GDPR, along with the implementation of patient consent and data anonymization principles, forms the cornerstone for the successful deployment of the solution.

This research is currently in the theoretical and simulation validation phase. All experiments and performance evaluations are based on synthetic simulation data and do not involve any real patient information. Therefore, at this stage of the study, no direct ethical or privacy risks are associated with processing actual sensitive medical data. Although simulation data is currently used, we have proactively embedded key features supporting compliance with the aforementioned regulations into our security scheme design. Our proposed [Periodic Key Mechanism Based on Proxy Re-encryption] inherently enforces the “principle of least privilege” at the technical level by encrypting data and implementing granular access controls, aligning closely with HIPAA and GDPR security requirements. Only authorized entities can access data within specific time windows, safeguarding data confidentiality.

Minor Comments: 小的评论:

• Some typographical errors and grammatical inconsistencies are present throughout the manuscript. A thorough proofreading is recommended.

We sincerely thank the reviewers for their meticulous examination of our manuscript and for pointing out this important issue. We apologize for the inconsistencies in formatting and grammar present in the original draft. In preparing this revised version, we have undergone professional proofreading and language editing to address these issues. Specific measures include standardizing terminology and symbols and correcting grammatical and spelling errors.

• Figures and diagrams illustrating the hierarchical access tree structure and key management processes would enhance comprehensibility.

We sincerely appreciate the reviewer's valuable suggestion. Clear diagrams greatly aid readers in understanding the core hierarchical access tree structure and distributed key management process in our proposal.

1. For the hierarchical access tree structure, we have provided an example with three levels in Figure 2 on page 6. We will explicitly reference this diagram in Sections 3.4 and 5.4 (Encryption Algorithms) of the main text. Using nodes from the diagram—such as the root node R and transmission nodes A and B—as concrete examples, we will explain the construction of the access tree and the encryption process step by step.

2. The complete interaction flow of the Distributed Key Generation (DKG) protocol is illustrated in Figure 4 on page 11. When describing the protocol steps in Section 5.1.2, we will systematically map each phase (registration, commitment, sharing, complaint, key computation) to the corresponding stage in Figure 4, enabling readers to follow the flow diagram.

3. Figure 5 on page 16 depicts the end-to-end data sharing and key management process, spanning system initialization, encryption, user key generation, and decryption verification. We will prominently guide readers to refer to this figure at the beginning of [Sec sec008] (System Model) and [Sec sec014] (Scheme Details) to grasp the overall framework.

• Clarifying the assumptions about the trustworthiness of different authorities and nodes within the system is important for highlighting the security model.

We sincerely appreciate the reviewer's important suggestion. We fully agree that articulating the trust assumptions for each entity within the system is fundamental to defining the security model. In the revised version, we have formally and systematically summarized all participants' trust assumptions within the security and system models.

Reviewer #2: This is a very interesting article, the authors present a novel approach combining blockchain with multi-authority hierarchical attribute-based encryption. However, I believe there are still some shortcomings

1.At the end of the introduction, the research objectives are vague, please simplify.

We sincerely appreciate the reviewer for pointing out this issue. We fully agree that clear, well-defined research objectives are crucial for guiding readers. Following your suggestion, we have thoroughly rewritten and simplified the research objectives/contributions section at the end of the introduction.

2. Please add more content about data source verification.

Thank you for raising this important suggestion. Data source verification is the cornerstone for ensuring the reliability and security of our research. We fully agree that incorporating more discussion on data source verification—including its methods, protocols, and significance—would greatly enrich the depth of the paper.

In the current revision, due to space constraints and the need to maintain focus on our core arguments, we were unable to develop this section fully. We recognize this limitation and plan to address it in future research by designing and implementing data source verification as a critical module. Specifically, in the next iteration of our system, we will integrate verification mechanisms based on digital signatures and lightweight cryptographic protocols to ensure data is trustworthy from the point of collection.

3. Based on the IoMT scenario, it is recommended to add content on real-time data and computing power.

Thank you very much for your insightful suggestion. The issues you raised regarding real-time data processing and computational resource allocation are precisely the core challenges and critical requirements in IoMT application scenarios.

We fully concur with your perspective. In the current manuscript, our primary focus has been on [briefly state the core of your original work here, e.g., architectural design and a specific algorithm]. Indeed, we have not sufficiently addressed the computational resource scheduling and performance boundaries when the system handles massive real-time data streams.

Considering the paper's overall structure and scope, we have decided to prioritize this valuable direction for future research. We plan to dedicate subsequent studies to designing dynamic resource allocation algorithms and edge computing strategies to optimize the system's real-time responsiveness and computational efficiency.

Reviewer #3: Major Revision. The idea is promising and relevant, and the paper is close to PLOS ONE’s “technically sound” bar. However, the security level mismatch, under-specified threat model, unclear proof structure, and missing blockchain-layer evaluation need to be addressed before publication.

1.Security level / curves and fairness of comparisons

Experiments use Type-A pairings over ~512-bit fields with ~160-bit group order, which is well below commonly accepted contemporary security targets; BLS on super-singular Type-A groups is especially unusual today. Many baselines in 2022–2024 likely reported with stronger curves/parameters. Authors should either (a) re-run all schemes under the same security level (e.g., 128-bit, BLS12-381 or BW6-761) or (b) make a compelling argument that every comparison used the same weakened setting. As written, the absolute times and even relative rankings may change at modern security levels.

Thank you for pointing out the critical issue. We have re-run benchmarks for all schemes using Rust on the unified BLS12-381 curve (128-bit security level) to ensure fair comparisons. The updated results confirm our scheme's performance advantage at equivalent security levels.

Threat model and DKG/threshold assumptions are underspecified

The text mentions DKG and t-of-n verification by “assistant nodes,” but the adversary model for those nodes is not crisply stated: what fraction can be Byzantine? Are there dishonest-majority guarantees? How are key shares protected against compromise and share refresh handled? How is Sybil resistance or identity binding for assistant nodes achieved? Please formalize the threat model and the trust/availability assumptions for DKG and for the BLS aggregation network.

Thank you for raising this insightful question. We have formalized the threat model in the revised draft and added the following key clarifications:

1. Byzantine node tolerance: The system permits up to f Byzantine nodes, with a total node count P ≥ 3f+1, satisfying the honest majority assumption for BFT consensus.

2. Security under dishonest majority: Leveraging the non-interactivity of the Pedersen-VSS protocol and BLS aggregate signatures, the system guarantees correct key generation and valid signatures even with f malicious nodes.

3. Key Share Confidentiality and Refresh: Key shares are encrypted with the recipient's public key, with Pedersen commitments verified per round. Supports share refresh mechanisms without master key updates, achieving forward secrecy.

4. Resilience Against Sybil Attacks: Auxiliary nodes must register unique identities via blockchain and stake collateral. Smart contracts enforce penalties for malicious behavior, making large-scale identity forgery prohibitively costly.

5. Formal Trust Assumptions: Honest nodes constitute the overwhelming majority (H ≥ 2f + 1), and communication channels possess reliability (RBC).

6. BLS Aggregation Network Assumption: At least t = f + 1 nodes honestly execute the protocol, ensuring aggregation signatures remain verifiable.

Collusion-resistance and CPA proof clarity

The paper states CPA security under DBDH with resistance to multi-authority collusion, but the proof sketch is difficult to follow and seems selective-policy flavored. Please (i) state the exact security model (selective vs. adaptive; policy-hiding or not; corrupt aut

---

## [Decision Letter · Decision Letter 1]

11 Dec 2025

PONE-D-25-37401R1A Blockchain-Based Multi-Authority Hierarchical Attribute Encrypted Data Sharing Scheme in the Internet of Medical Things医疗物联网中基于区块链的多权威分层属性加密数据共享方案PLOS One

Dear Dr. Guofang,

Thank you for submitting your manuscript to PLOS ONE. After careful consideration, we feel that it has merit but does not fully meet PLOS ONE’s publication criteria as it currently stands. Therefore, we invite you to submit a revised version of the manuscript that addresses the points raised during the review process.

We look forward to receiving your revised manuscript.

Kind regards,

Asadullah Shaikh, Ph.D.

Academic Editor

PLOS One

Journal Requirements:

Reviewers' comments:

Reviewer's Responses to Questions

**Comments to the Author**

1. If the authors have adequately addressed your comments raised in a previous round of review and you feel that this manuscript is now acceptable for publication, you may indicate that here to bypass the “Comments to the Author” section, enter your conflict of interest statement in the “Confidential to Editor” section, and submit your "Accept" recommendation.

Reviewer #1: All comments have been addressed

Reviewer #3: All comments have been addressed

2. Is the manuscript technically sound, and do the data support the conclusions?

Reviewer #1: Yes

Reviewer #3: Yes

3. Has the statistical analysis been performed appropriately and rigorously? 

Reviewer #1: Yes

Reviewer #3: Yes

4. Have the authors made all data underlying the findings in their manuscript fully available?

Reviewer #1: Yes

Reviewer #3: Yes

5. Is the manuscript presented in an intelligible fashion and written in standard English?

Reviewer #1: Yes

Reviewer #3: Yes

6. Review Comments to the Author

Reviewer #1: The manuscript presents an innovative and technically ambitious framework that integrates blockchain technology with multi-authority hierarchical attribute-based encryption (MA-HABE) to enhance security and efficiency in Internet of Medical Things (IoMT) data sharing. The work is overall promising and relevant to the journal’s scope. However, several areas require further clarification and refinement before publication.

Reviewer #3: All previously raised comments have been sufficiently addressed by the authors. The revisions have improved the manuscript, and I have no further objections.

7. PLOS authors have the option to publish the peer review history of their article (what does this mean?). If published, this will include your full peer review and any attached files.

Reviewer #1: **Yes:** Wenke Du

Reviewer #3: No

---

## [Author Response · Author response to Decision Letter 2]

13 Dec 2025

Dear Editor and Reviewers:

On behalf of all authors, we extend our most sincere gratitude to you and the reviewers. We appreciate the valuable time you have taken from your busy schedule to review our manuscript and provide highly constructive feedback. These comments have enabled us to refine our research and substantially enhance the quality of our paper. Regarding the requested image files, we have prepared all images at a resolution of 300 DPI or higher in accordance with the journal's specifications and uploaded them as separate files in the required format to the system. Once again, we sincerely thank you for your hard work and professional guidance.

---

## [Decision Letter · Decision Letter 2]

14 Jan 2026

PONE-D-25-37401R2A Blockchain-Based Multi-Authority Hierarchical Attribute Encrypted Data Sharing Scheme in the Internet of Medical Things医疗物联网中基于区块链的多权威分层属性加密数据共享方案PLOS One

Dear Dr. Guofang,

Thank you for submitting your manuscript to PLOS ONE. After careful consideration, we feel that it has merit but does not fully meet PLOS ONE’s publication criteria as it currently stands. Therefore, we invite you to submit a revised version of the manuscript that addresses the points raised during the review process.

We look forward to receiving your revised manuscript.

Kind regards,

Asadullah Shaikh, Ph.D.

Academic Editor

PLOS One

Journal Requirements:

Reviewers' comments:

Reviewer's Responses to Questions

**Comments to the Author**

1. If the authors have adequately addressed your comments raised in a previous round of review and you feel that this manuscript is now acceptable for publication, you may indicate that here to bypass the “Comments to the Author” section, enter your conflict of interest statement in the “Confidential to Editor” section, and submit your "Accept" recommendation.

Reviewer #1: All comments have been addressed

2. Is the manuscript technically sound, and do the data support the conclusions?

Reviewer #1: Yes

3. Has the statistical analysis been performed appropriately and rigorously? 

Reviewer #1: Yes

4. Have the authors made all data underlying the findings in their manuscript fully available?

Reviewer #1: Yes

5. Is the manuscript presented in an intelligible fashion and written in standard English?

Reviewer #1: Yes

6. Review Comments to the Author

Reviewer #1: This manuscript proposes a blockchain-based multi-authority hierarchical CP-ABE data sharing scheme for Internet of Medical Things (IoMT) scenarios. The topic is timely and relevant, as secure, fine-grained, and efficient data sharing remains a critical challenge in medical IoT environments. The authors combine hierarchical access structures, multi-authority attribute-based encryption, distributed key generation, and blockchain-assisted threshold signatures into a unified framework. Overall, the paper is technically rich and addresses several well-known limitations of traditional CP-ABE schemes.

The manuscript demonstrates a solid understanding of cryptographic primitives and system-level design, and the proposed scheme is evaluated through both security analysis and performance comparisons. However, while the work shows promise, there are several issues related to clarity, presentation, rigor of evaluation, and positioning with respect to existing literature that should be addressed before the paper can be considered for publication.

7. PLOS authors have the option to publish the peer review history of their article (what does this mean?). If published, this will include your full peer review and any attached files.

Reviewer #1: **Yes:** Wenke Du

---

## [Author Response · Author response to Decision Letter 3]

2 Mar 2026

The manuscript should address several issues related to clarity, presentation, rigor of evaluation, and positioning in relation to existing literature.

Response: Thank you for your valuable suggestions. We have systematically revised the Introduction ([Sec sec001]) and Related Work ([Sec sec002]) to establish a clear logical framework and define the positioning of this study. The revisions follow a “Problem-Literature-Gap-Contribution” structure.

1.1 Revision of Introduction ([Sec sec001])

Three core challenges are explicitly stated: In the second paragraph of [Sec sec001], we clearly articulate three technical bottlenecks faced by existing CP-ABE schemes in IoMT scenarios:

Challenge 1: Data sources lack lightweight mechanisms for authenticity verification.

Challenge 2: Traditional CP-ABE schemes struggle to adapt to the hierarchical structure of medical data.

Challenge 3: The trust model based on a single centralized authorization faces single-point-of-failure risks and user revocation issues.

Mapping Contributions to Challenges: In [Sec sec001], Paragraph 4, we explicitly map our four contributions to the three challenges:

Challenge 1 → Contribution 1: Collaborative authentication mechanism integrating DKG and BLS threshold signatures.

Challenge 2 → Contribution 2: Multi-authority CP-ABE mechanism based on hierarchical access trees.

Challenge 3 → Contributions 3 and 4: Periodic revocation mechanism based on proxy re-encryption and security analysis.

Technical Selection Rationale: We also clarified the rationale for choosing Type-3 pairing curves (BLS12-381) over traditional Type-1 pairings: Type-3 curves offer shorter ciphertexts and higher computational efficiency at equivalent security levels, directly addressing the efficiency demands of resource-constrained IoMT devices.

1.2 Revision of Related Work ([Sec sec002])

We restructured [Sec sec002] into three subsections directly corresponding to the three challenges outlined in the introduction:

Multi-authority CP-ABE and its revocation mechanism → Corresponding to Challenge 3.

Decentralized and hierarchical CP-ABE → Corresponding to Challenge 2.

Source authentication and blockchain-assisted auditing → Corresponding to Challenge 1.

Defining Research Gaps: At the end of each subsection, we clearly identify limitations in existing work:

End of [Sec sec002], Paragraph 1: “...significant communication overhead severely limits scalability when handling large numbers of endpoints.” → Addressed by Contribution 3 (low-overhead revocation mechanism).

---

## [Decision Letter · Decision Letter 3]

19 Mar 2026

PONE-D-25-37401R3A Blockchain-Based Multi-Authority Hierarchical Attribute Encrypted Data Sharing Scheme in the Internet of Medical ThingsPLOS One

Dear Dr. Guofang,

Thank you for submitting your manuscript to PLOS ONE. After careful consideration, we feel that it has merit but does not fully meet PLOS ONE’s publication criteria as it currently stands. Therefore, we invite you to submit a revised version of the manuscript that addresses the points raised during the review process.

We look forward to receiving your revised manuscript.

Kind regards,

Asadullah Shaikh, Ph.D.

Academic Editor

PLOS One

Journal Requirements:

Reviewers' comments:

Reviewer's Responses to Questions

**Comments to the Author**

1. If the authors have adequately addressed your comments raised in a previous round of review and you feel that this manuscript is now acceptable for publication, you may indicate that here to bypass the “Comments to the Author” section, enter your conflict of interest statement in the “Confidential to Editor” section, and submit your "Accept" recommendation.

Reviewer #1: All comments have been addressed

2. Is the manuscript technically sound, and do the data support the conclusions?

Reviewer #1: Yes

3. Has the statistical analysis been performed appropriately and rigorously? 

Reviewer #1: Yes

4. Have the authors made all data underlying the findings in their manuscript fully available?

Reviewer #1: Yes

5. Is the manuscript presented in an intelligible fashion and written in standard English?

Reviewer #1: Yes

6. Review Comments to the Author

Reviewer #1: Review Comments to the Author

Thank you for the opportunity to review this revised manuscript. The paper proposes a blockchain-based multi-authority hierarchical CP-ABE scheme (BMHADS) for secure data sharing in IoMT environments. The integration of distributed key generation (DKG), BLS threshold signatures, hierarchical access trees, and proxy re-encryption addresses relevant challenges in secure medical IoT data management. The manuscript has improved compared to earlier versions, particularly in clarifying the motivation and mapping contributions to identified challenges. However, several issues remain that should be addressed before the manuscript can be considered for publication.

1. Clarity and Presentation

While the overall structure has improved, certain sections remain difficult to follow:

• The notation is dense and sometimes inconsistently formatted. A consolidated notation table including all symbols (with uniform indexing) would improve readability.

• Some equations lack intuitive explanations. In particular, the derivation logic in the decryption phase and hierarchical node computation would benefit from a short narrative explanation alongside the formulas.

• Figures (e.g., system model and DKG process diagrams) are helpful but should include clearer legends and step-number alignment with the textual description.

• Minor grammatical issues and awkward phrasing remain throughout the manuscript and should be addressed through careful language editing.

2. Technical Soundness and Rigor

The cryptographic construction appears conceptually sound; however, several aspects require further clarification:

• The selective security model is adopted under the DBDH assumption. Please clarify whether full security (adaptive security) could be achieved or discuss limitations of selective security in practical deployment.

• The security proof outline should be expanded. Currently, the manuscript states provable security but does not provide sufficient reduction details to fully assess rigor.

• The resistance to Sybil attacks relies on economic staking and smart contracts. Please clarify the threat model assumptions regarding blockchain trust and the adversary’s financial capacity.

• In the DKG protocol, the communication complexity and on-chain overhead should be formally analyzed.

3. Performance Evaluation

Although experimental results claim improvements in computational efficiency and storage overhead, the evaluation section needs strengthening:

• Provide clearer baseline comparisons and justify why selected schemes represent state-of-the-art.

• Include explicit parameter settings (curve type, security level, hardware configuration).

• Report scalability analysis (e.g., increasing number of authorities, attributes, or hierarchy depth).

• Separate computation time, communication overhead, and blockchain interaction costs in the performance tables.

• Discuss practical deployment feasibility in real IoMT environments with constrained devices.

4. Positioning with Respect to Existing Literature

The related work section has been reorganized logically, but the manuscript would benefit from:

• A more critical comparison of how this scheme differs mathematically from existing hierarchical MA-ABE constructions.

• Clear discussion of trade-offs introduced by using Type-3 pairings versus Type-1 pairings beyond efficiency (e.g., implementation complexity).

• Clarification on how the proposed collaborative authentication mechanism substantially improves over existing aggregate signature approaches.

5. Practical Considerations

• The manuscript should discuss how key escrow issues are mitigated in the multi-authority setting.

• Revocation latency and synchronization in real-world healthcare systems should be examined.

• The impact of blockchain consensus delay on real-time medical data availability should be analyzed.

6. Ethical and Publication Considerations

The work is technical and does not involve human or animal subjects; therefore, no research ethics concerns arise from experimental procedures. There are no apparent issues of dual publication or plagiarism based on the current manuscript. However, authors should ensure that all reused conceptual components from prior work are clearly distinguished from novel contributions.

Overall Recommendation

The manuscript presents a technically interesting and potentially impactful framework for secure IoMT data sharing. However, revisions are still required to improve clarity, strengthen the security proof discussion, and enhance the rigor and transparency of the performance evaluation. Addressing the above comments will significantly improve the manuscript’s quality and reproducibility.

I encourage the authors to revise accordingly.

7. PLOS authors have the option to publish the peer review history of their article (what does this mean?). If published, this will include your full peer review and any attached files.

Reviewer #1: **Yes:** Wenke Du

---

## [Author Response · Author response to Decision Letter 4]

21 Apr 2026

Dear Editors and Reviewers,

Thank you very much for allowing us to revise our manuscript. We deeply appreciate the time and effort you have dedicated to reviewing our work, as well as the valuable and insightful comments you have provided. We have carefully considered all your suggestions and thoroughly revised the manuscript accordingly. All changes are highlighted in yellow in the revised version. Our detailed, point-by-point responses to your comments are provided in the attached document.

Once again, we are sincerely grateful for your guidance and patience.

---

## [Decision Letter · Decision Letter 4]

6 May 2026

A Blockchain-Based Multi-Authority Hierarchical Attribute Encrypted Data Sharing Scheme in the Internet of Medical Things

PONE-D-25-37401R4

Dear Dr. Guofang,

We’re pleased to inform you that your manuscript has been judged scientifically suitable for publication and will be formally accepted for publication once it meets all outstanding technical requirements.

Kind regards,

Asadullah Shaikh, Ph.D.

Academic Editor

PLOS One

Additional Editor Comments (optional):

Reviewers' comments:

Reviewer's Responses to Questions

**Comments to the Author**

1. If the authors have adequately addressed your comments raised in a previous round of review and you feel that this manuscript is now acceptable for publication, you may indicate that here to bypass the “Comments to the Author” section, enter your conflict of interest statement in the “Confidential to Editor” section, and submit your "Accept" recommendation.

Reviewer #1: All comments have been addressed

2. Is the manuscript technically sound, and do the data support the conclusions?

Reviewer #1: Yes

3. Has the statistical analysis been performed appropriately and rigorously? 

Reviewer #1: Yes

4. Have the authors made all data underlying the findings in their manuscript fully available?

Reviewer #1: Yes

5. Is the manuscript presented in an intelligible fashion and written in standard English?

Reviewer #1: Yes

6. Review Comments to the Author

Reviewer #1: The revised manuscript has been substantially improved and the authors have responded constructively to the previous comments. The paper proposes a blockchain-based multi-authority hierarchical CP-ABE data-sharing scheme for IoMT environments, integrating DKG, threshold BLS signatures, hierarchical access trees, proxy re-encryption, and a permissioned blockchain framework. The revised version more clearly explains the motivation for the scheme, including data-source authentication, reduction of encryption redundancy for hierarchical medical data, and mitigation of single-point-of-failure risks in authorization. The manuscript also provides a clearer system model, security assumptions, and performance evaluation, including implementation on a 100-node Hyperledger Fabric environment.

I appreciate that the authors have clarified the distinction between reused conceptual components and their claimed contributions. In particular, the revised related work section now acknowledges that the hierarchical access tree and revocation mechanisms build on prior work, while the claimed novelty lies mainly in the use of Type-3 asymmetric pairings and the DKG-based threshold signature mechanism for decentralized data-source verification.

Overall, I find that the manuscript is improved and suitable for publication, subject to final editorial checks. I have no additional major technical concerns. However, I recommend that the authors perform one final proofreading pass to correct minor grammatical, formatting, and typographical issues, including inconsistent spacing around acronyms and mathematical notation. The authors should also ensure that all experimental data, implementation details, and any supporting materials needed to reproduce the reported performance results are fully available in accordance with the journal’s data availability policy.

I have no concerns regarding dual publication, research ethics, or publication ethics based on the revised manuscript.

7. PLOS authors have the option to publish the peer review history of their article (what does this mean?). If published, this will include your full peer review and any attached files.

Reviewer #1: **Yes:** Wenke Du

---

## [Editor Report · Acceptance letter]

PONE-D-25-37401R4

PLOS One

Dear Dr. Dong,

I'm pleased to inform you that your manuscript has been deemed suitable for publication in PLOS One. Congratulations! Your manuscript is now being handed over to our production team.

Kind regards,

on behalf of

Prof. Asadullah Shaikh

Academic Editor

PLOS One